# GenColorBench: A Color Evaluation Benchmark for Text-to-Image Generation Models

## Abstract

Recent years have seen impressive advances in text-to-image generation, with image generative or unified models, generating high-quality images from text. Yet these models still struggle with fine-grained color controllability, often failing to accurately match colors specified in text prompts. While existing benchmarks evaluate compositional reasoning and prompt adherence, none systematically assess the color precision. Color is fundamental to human visual perception and communication, critical for applications from art to design workflows requiring brand consistency. However, current benchmarks either neglect color or rely on coarse assessments, missing key capabilities like interpreting RGB values or aligning with human expectations. To this end, we propose GenColorBench, the first comprehensive benchmark for T2I color generation, grounded in color systems like ISCC-NBS and CSS3/X11, including numerical colors which are absent elsewhere. With 44K color-focused prompts covering 400+ colors, it reveals models' true capabilities via perceptual and automated assessments. Evaluations of popular T2I models using GenColorBench show performance variations, highlighting which color conventions models understand best and identifying failure modes. Our GenColorBench assessments will allow to guide improvements in precise color generation. The benchmark will be made public upon acceptance.

## 1 Introduction

Text-to-image (T2I) generation has witnessed remarkable progress in recent years, with state-of-the-art models like Stable Diffusion (Rombach et al., 2022) and FLUX (Labs, 2024) demonstrating unprecedented capabilities in generating high-quality, photorealistic images from text prompts. These advances have enabled diverse applications ranging from creative content generation to automated design workflows. However, despite their impressive overall performance, T2I models still struggle with fine-grained controllability, particularly in generating images that precisely match specific visual attributes described in text prompts (Chefer et al., 2023; Ge et al., 2023a). While numerous benchmarks, discussed in Table 1, have been proposed to evaluate various aspects of T2I model performance—including compositional reasoning (Huang et al., 2025; Ghosh et al., 2023), prompt adherence (Hu et al., 2024), and faithfulness (Hu et al., 2023)—none systematically evaluates the critical ability to generate precise colors as specified in text prompts.

Color represents a fundamental dimension of human visual perception and serves as a primary channel for human communication about objects and scenes, with color categories forming a universal basis for describing and distinguishing visual phenomena across cultures (Berlin & Kay, 1991; Witzel & Gegenfurtner, 2018). This perceptual importance translates directly into practical applications where accurate color generation is essential—from multimedia applications and artistic creation to design workflows requiring brand consistency, aesthetic control and faithful reproduction of real-world scenes. However, existing T2I evaluation benchmarks critically underestimate this importance by either neglecting color evaluation entirely or reducing it to coarse categorical assessments that fail to capture their real color capabilities. Current benchmarks do not assess whether models generate colors that maintain color consistency across different contexts, or produce colors that align with human memory and expectations for familiar objects.

To address this, we propose GenColorBench, the first comprehensive benchmark designed to systematically evaluate the color generation capabilities of T2I models. Unlike existing benchmarks

| Benchmark | Scale | Focus | Color Evaluation Tasks | | | | | Color Evaluation Methods |
|---|---|---|---|---|---|---|---|---|
| | | | CNA | MCC | COA | NCU | ICA | |
| GenEval (Ghosh et al., 2023) | 553 | Compositionality | ✓ | ✓ | ≈ | × | × | Mask2Former + CLIP ViT-L/14 |
| T2I-CompBench++ (Huang et al., 2025) | 6000 | Compositionality | ✓ | ≈ | × | × | × | BLIP-VQA |
| DPG-Bench (Hu et al., 2024) | 1065 | Prompt Adherence | ✓ | × | × | × | × | mPLUG-large VQA |
| TIFA (Hu et al., 2023) | 1000 | Faithfulness | ✓ | ✓ | × | × | × | mPLUG-large VQA |
| Commonsense-T2I (Fu et al., 2024) | 1000+ | Reasoning | ≈ | ≈ | × | × | × | self-proposed (accuracy) |
| Winoground-T2I (Zhu et al., 2023) | 11,000 | Compositionality | ✓ | ✓ | × | × | × | Human Rating + DSG-VQA |
| Wise (Niu et al., 2025) | 1000 | Reasoning | ≈ | ≈ | × | × | × | WiScore, Aesthetic Quality |
| MMMG (Luo et al., 2025) | 4456 | Disciplinary Knowledge | ✓ | ✓ | ✓ | × | × | GPT/Gemini/QWEN VQA |
| Partiprompt (Yu et al., 2022) | 1600 | Compositionality | ✓ | ✓ | × | × | × | FID |
| OneIG-Bench (Chang et al., 2025) | 2440 | Compositionality | × | × | × | × | × | FID |
| DrawBench (Saharia et al., 2022) | 200 | Compositionality | ✓ | × | × | × | × | Human Rating |
| EvalAlign (Tan et al., 2024) | 3000 | Compositionality | ✓ | ✓ | × | × | × | MLLM-VQA |
| Evalmuse (Han et al., 2024) | 4000 | Compositionality | ✓ | ✓ | × | × | × | FGA-BLIP2, PN-VQA |
| **GenColorBench (Ours)** | 44,464 | Color Understanding | ✓ | ✓ | ✓ | ✓ | ✓ | VQA + Color Metrics |
| **GenColorBench-Mini (Ours)** | < 10K | Color Understanding | ✓ | ✓ | ✓ | ✓ | ✓ | VQA + Color Metrics |

Table 1: Overview of existing T2I evaluation benchmarks. Abbreviations for color evaluation tasks: CN = Color Name Understanding, MC = Multi-Color Composition, CO = Color–Object Association, NCU = Numeric Color Understanding, ICA = Implicit Color Association. *While these benchmarks are widely adopted for assessing various aspects of T2I generation—such as compositionality, prompt adherence, and reasoning—they lack comprehensive coverage of key color understanding and evaluation tasks.* **GenColorBench** *is specifically designed to fill this gap by supporting a broad spectrum of color-related tasks.* (✓: covered, ×: not covered, ≈: partially covered)

that rely on coarse categorical assessments, our benchmark is grounded in established color naming systems, including the ISCC-NBS, and CSS3/X11, and uniquely incorporates evaluation of numerical color specifications (RGB values and hex codes) that are completely absent from existing benchmarks. With over 44K+ prompts specifically designed for color evaluation covering over 400+ colors, GenColorBench provides both the scale and specificity necessary to reveal models' true color generation capabilities through both perceptual color evaluation and automated assessment methods.

We conduct extensive evaluations of several popular image generation models and unified models using GenColorBench, revealing significant variations in color generation capabilities across different models and color specification methods. Our analysis provides insights into which color naming conventions and numerical representations are most effectively understood by current models, and identifies common failure modes in color generation tasks. The main contributions of this work are threefold: (i) We introduce GenColorBench, a large-scale benchmark containing over 44,464 prompts covering 400+ colors specifically designed to evaluate the capabilities of T2I models across five distinct color generation tasks; (ii) We provide comprehensive evaluations of state-of-the-art T2I models, analyzing their performance on precise color generation and identifying key limitations; (iii) We establish baseline performance metrics and evaluation protocols that can guide future research in improving color controllability in generative models.

---

**MAJOR FINDINGS OF OUR EVALUATION**

- Current models exhibit notable shortcomings in adhering to precise color specifications, underscoring the urgent need for enhanced color controllability (Table 4).
- Model performance is tightly linked to category semantics. Categories with strong color associations (e.g. Fruits and Vegetables—yellow bananas, green grass) pose greater challenges (Fig. 2, Fig. 3).
- Models are better at understanding basic colors (yellow, pink, blue), while they struggle more with intermediate colors (Fig. 5(Left)). Similarly, models favor "light" and "dark" modifiers over more nuanced ones like "-ish", suggesting a limited grasp of subtle color variations (Fig. 5(Right)).
- Vision-language models fall short as reliable tools for color evaluation (Table 2).

---

## 2 RELATED WORK

**T2I Diffusion Models.** T2I generation has advanced rapidly in recent years. T2I diffusion models (Ho et al., 2020; Gu et al., 2022) emerged as more efficient models surpassing GANs (Goodfellow et al., 2020), VAEs (Kingma & Welling, 2013), autoregressive (Esser et al., 2021) and flow-

based (Dinh et al., 2015; 2017) models in T2I generation. Diffusion models are probabilistic generative models aiming to learn data distribution through denoising from Gaussian distribution. These models allow multi-modal conditioning (Song et al., 2021), (Meng et al., 2022), (Nichol et al., 2021) to improve controllability. With recent scaling up the scale of diffusion models, SD3 (Esser et al., 2024) and FLUX (Labs, 2024) have been state-of-the-art T2I models while largely surpassing the previous representatives (Ramesh et al., 2022; Chen et al., 2023).

**Unified Models.** Recent years have seen major progress in multimodal understanding and image generation models. Yet, these fields have advanced along separate paths, forming distinct architectural paradigms. Autoregressive architectures dominate large language models such as LLaMa (Touvron et al., 2023), Qwen (Team, 2024a), and multimodal models like LLaVa (Liu et al., 2023), Qwen-VL (Team, 2024b). Autoregressive-based architectures have established dominance in large language models such as LLaMa (Touvron et al., 2023), Qwen (Team, 2024a), etc, as well as in multimodal understanding models including LLaVa (Liu et al., 2023) and Qwen-VL (Team, 2024b). Diffusion models, such as Stable Diffusion (Podell et al., 2023) and FLUX (Labs, 2024), have become central to image generation, producing high-fidelity, prompt-aligned images. More recently, unified frameworks like GPT-4o aim to handle multimodal inputs and outputs in a single mechanism. Unified models fall into three types: diffusion-based, autoregressive (AR), and fused AR/diffusion. Pure diffusion-based MLLMs, such as MMaDA (Yang et al., 2025) and Dual-Diffusion, use dual-branch diffusion for joint text–image generation. However, unified models based on naive autoregressive (AR) dominate this research landscape, with representative contributions including SEED series (Ge et al., 2023b), Emu series (Sun et al., 2024), Janus series (Wu et al., 2025a; Chen et al., 2025b), etc. Recently, fused AR–diffusion models have emerged for unified vision–language generation, exemplified by Show-o (Xie et al., 2024b) and BAGEL (Deng et al., 2025).

**Color Control in T2I diffusion models.** With the advancements in generation and unified models, various text-guided image editing approaches (Hertz et al., 2023a; Meng et al., 2022; Mokady et al., 2023) have been developed to enable controllable modifications. For instance, methods like Imagic (Kawar et al., 2023) and P2P (Hertz et al., 2023b) leverage Stable Diffusion (SD) models for structure-preserving edits. And the unified models (Deng et al., 2025; Wu et al., 2025b) integrate such editing power by large-scale pretraining with huge paired datasets. Another technique stream which can also achieve controllable generation is transfer learning for T2I models (Ruiz et al., 2023; Kumari et al., 2023). It aims at adapting a given model to a *new concept* by given images from the users and bind the new concept with a unique token. As a result, the adaptation model can generate various renditions for the new concept guided by text prompts. However, all these existing techniques struggle to achieve fine-grained control over color attributes in image editing and generation tasks. Only a limited number of works (Butt et al., 2024; Ge et al., 2023a) have begun addressing the challenge of precise color generation. To facilitate the evaluation and development of precise color generation capabilities of future models, we build the first color benchmark in this paper.

**T2I Evaluation.** A variety of benchmarks have been developed to evaluate text-to-image models, each tailored to specific aspects of generative performance, as listed in Table 1. GenEval (Ghosh et al., 2023) introduces object detectors to enable fine-grained, object-level evaluation, thereby addressing the limitations of holistic metrics. T2I-CompBench (Huang et al., 2025) elevates compositional complexity by constructing prompts that integrate attributes, relational cues, numeracy, and complex scene descriptions. DPG-Bench (Hu et al., 2024) focuses on assessing models' instruction-following proficiency, leveraging text-rich prompts to gauge their fidelity to detailed directives. Furthermore, Commonsense-T2I (Fu et al., 2024) employs adversarial prompts to probe models' capabilities in visual reasoning. Winoground-T2I (Zhu et al., 2023) evaluates compositional generalization by leveraging contrastive sentence pairs. More recently, WISE (Niu et al., 2025) and MMMG (Luo et al., 2025) benchmarks emphasize world knowledge-based evaluation, spanning cultural, scientific, and temporal domains to gauge models' alignment with broader understanding. However, these existing benchmarks are primarily designed to evaluate the general generative capabilities of diverse image generators, with none specifically focusing on the task of color generation.

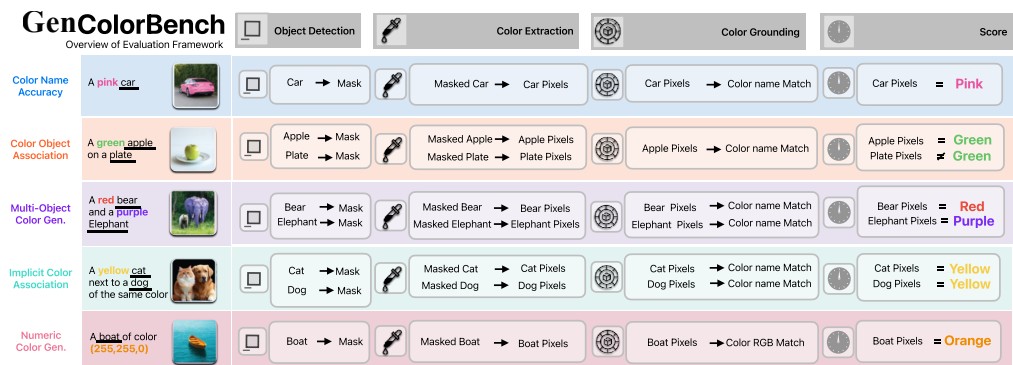

Figure 1: An overview of GenColorBench evaluation framework. The evaluation pipeline consists of five key components: VQA-based object localization, object segmentation, pixel extraction, color grounding, and score mechanism. Then, five color evaluation tasks are devised to analyse different aspects of color understanding in T2I models covering single object coloring, color-object association, multi-object color composition, numerical color understanding, and Implicit Color Association.

## 3 COLOR EVALUATION FRAMEWORK

### 3.1 T2I COLOR GENERATION TASKS.

Our primary goal is to evaluate unified vision-language and T2I models' ability to understand and generate images given explicit color prompts. We organize evaluation into multiple tasks, targeting different dimensions of color understanding, considering the practical use-cases for generative models. GenColorBench consists of five color evaluation tasks: (i) **Color Name Accuracy**—assesses whether the model correctly renders an object in the color specified by its linguistic name. (ii) **Color-Object Association**—evaluates whether the specified color is assigned to the correct object without erroneous attribution to contextual elements. (iii) **Multi-Object Color Composition**—assess correct color-object associations when multiple objects and corresponding color names are specified. (iv) **Implicit Color Association**—evaluates understanding of semantic relationships when a color is assigned to only one object but should also correspond to other objects. (v) **Numerical Color Understanding**—assesses comprehension of RGB triplets and hex codes for accurate color generation.

### 3.2 COLOR TAXONOMY

Colors can be specified in text prompts in various ways—most commonly through linguistic color names such as "a red rose", but also through numerical codes such as hexadecimals (e.g., `#ff0000`) or RGBs (e.g., (255, 0, 0)). These color expressions are often interpreted differently by the T2I models depending on their text encoders. Therefore, it is important to consider both the linguistic and numerical color representations to perform an in-depth evaluation of T2I models for color generation tasks. To this end, we ground our evaluation in two standard color naming systems i.e., ISCC-NBS, and CSS/X11 which offers human-understandable names along with their numerical representations.

The ISCC-NBS (Kelly & Judd, 1976) is derived from the Munsell color system (Munsell, 2022) that is a perceptually uniform color space designed to align with the human color perception. Munsell's color system organizes colors along three perceptual axes, which are hue, value (lightness), and chroma (saturation), determined by empirical human experiments. ISCC-NBS discretizes this continuous color space into named categories, resulting in a three-level hierarchy of colors, ranging from coarse to fine-grained colors. Level 1 includes 13 broad color categories corresponding to basic color linguistic names such as green, red, or blue. Level 2 expands these 13 colors to 29 intermediate hues by incorporating modifiers such as light, deep, or strong. Level 3 provides fine-grained color names with precise distinctions, such as light bluish green or moderate purplish pink. We also use CSS3/X11 color set (W3C, 2018), which includes 147 colors that are widely used in web design and digital interfaces. These color names precisely map to both RGB and hexadecimal color values, making them ideal to be used in text-prompts for T2I color generation evaluation tasks.

| Model | Open-Ended | | MCQ | | Binary | |
|---|---|---|---|---|---|---|
| | CSS | L2 | CSS | L2 | CSS | L2 |
| Janus 1.3B | 5.03 | 25.86 | 12.20 | 33.99 | 30.42 | 34.98 |
| Janus-Pro 7B | 6.62 | 26.60 | 19.44 | 43.60 | 24.98 | 37.19 |
| mPLUG-Owl3 7B | 7.24 | 24.14 | 17.93 | 42.12 | 26.87 | 41.87 |
| DeepSeek-VL2-7B | 11.35 | 27.34 | 18.85 | 45.32 | 31.24 | 42.12 |
| BLIP3o-8B | 12.17 | 25.12 | 24.73 | 45.81 | 31.10 | 44.09 |
| Qwen2-VL-7B | 9.35 | 24.63 | 23.23 | 43.35 | 35.13 | 49.01 |
| Instruct-VL-7B | 7.19 | 26.60 | 20.55 | 45.57 | 31.15 | 41.63 |
| Ours | **L2: 96.46** | | | | **CSS: 92.00** | |

Table 2: Performance (accuracy) of VLMs-based VQA on CSS/X11 and ISCC-NBS Level 2 colors.

| Type | # Temp. | Example Prompt |
|---|---|---|
| Object-Focused | 12 | *a red apple* |
| Contextual Object | 62 | *a red apple on a white plate* |
| Scene Descriptive | 30 | *a red apple on a white plate placed on a kitchen shelf* |
| Implicit Color Association | 100 | *a red apple on a plate placed on a kitchen shelf. The plate is of the same color as the apple.* |

Table 3: Prompt categorization across four levels of difficulty, from simple to complex.

### 3.3 DATA CURATION

After establishing the color evaluation tasks and the color sets, we generate prompts for each color evaluation task. The data curation involves four key components: object selection, prompt template creation and categorization, integration of standardized colors, and human-in-the-loop quality assessment. Each component is designed to ensure that the generated prompts and the associated evaluation settings are grounded, scalable, and suitable for automated and human evaluation.

**Object Selection.** We curate a set of 108 objects that span multiple semantic categories to ensure comprehensive coverage of color-object combinations. These objects are drawn from two widely used datasets—COCO (Lin et al., 2014), and ImageNet (Deng et al., 2009), and grouped them into seven semantic domain including fruits and vegetables, tools and miscellaneous items, vehicles, animals, clothing and accessories, furniture and household objects, and sports and toys. Each object is selected based on recognizability in T2I generation, color variability for plausible appearance, and suitability for the segmentation which is a crucial step in the downstream mask-based evaluation.

**Prompt Creation and Categorization.** We begin by pairing the objects and the color sets, resulting in a large pool of valid object-color combinations that serves as a seed inputs for the prompt generation. For each color-object pair, we use a pool of hand-crafted and GPT-4o generated prompt templates to produce the prompts, which are aligned with one of the four difficulty levels—shown in Table 3. Level 1 templates produce simple object focused prompts that describe a single colored object. These prompts are designed to evaluate the color name accuracy and numerical color understanding task. Level 2 templates embed the object within a contextual scene which are used for color name accuracy and color-object association task. Level 3 templates describe the scene involving more than two objects along with their corresponding colors to assess the multi-object color compositions. Level 4 templates describe semantically complex scenes having one object with the assigned color, while a second object is referring to the color of the first object.

**Quality Assessment.** After completing prompt generation, we perform human-in-the-loop validation to ensure the linguistic quality and semantic clarity of the generated prompts. The prompts are reviewed for grammatical check, and ambiguity, especially in scene descriptive and implicit color association prompts. A random subset of prompts from each set are picked for review to ensure that the color references are unambiguous and the prompt structure does not mislead the models. All the ambiguous prompts are either revised or removed from the final sets.

**Prompt Distribution.** Finally, we get 18K object focused prompts with linguistic color names, and 11.5K prompts with numerical colors including hex codes and RGB triples. The contextual object category includes 8.7K prompts to assess the object-color association. To evaluate multiple object generation, the scene descriptive category contains 2.2K prompts that embed colors within broader contexts. The implicit color association category includes 4.5K prompts where color attributes must be inferred based on semantic relationships between objects. This prompt distribution ensures a comprehensive evaluation of color grounding across a wide range of complexity levels, resulting into a large-scale set of 44K+ prompts. To facilitate broader accessibility and reproducibility, we further curate a compact, representative subset of less than 10K prompts—carefully selected to preserve semantic diversity and evaluation fidelity—making it readily usable by the research community.

### 3.4 EVALUATION FRAMEWORK

**Object Detection.** Our framework comprises three key components: object detection and segmentation, color grounding, and scoring mechanism to ensure object-aware perceptually aligned assess-

ment. Following the Davidsonian Scene Graph (DSG) framework (Cho et al., 2023), we employ Visual Question Answering (VQA)-based validation to first confirm the presence of the intended object(s) in the generated image before proceeding to attribute-level assessments such as color. For instance, given an input image along with ground truth, we formulate binary queries such as "Is there a car in the image", and rely on VQA response to determine the existence of object. For the multi-object tasks, the VQA model is queried for each object separately, and the image is validated only if all the objects in text prompts are present in the image. This ensures object-level precision in the evaluation tasks, especially in those that involve color association and color grounding between multiple objects. In practice, after empirical testing across several VLMs, we employ Janus-1.3B as VQA model due to its favorable trade-off between computational efficiency and reliability.

Then, a binary mask of the object is generated for color extraction. We use Grounded SAM (Ren et al., 2024) pipeline which uses grounding DINO for text guided coarse localization of object and then SAM is used to produce final mask. Another reason for employing Grounded SAM is that the object may contain additional associated regions not required for the color grounding i.e., a mask of car may include lights, and wind shields that are not required in the color grounding. We refer these components as negative labels, and generated a list of the negative labels for all the objects using GPT-4o. To remove these negative objects from the mask, we apply negative Intersection-over-Union (IoU) filtering over positive mask to ensure separation of spatial region of the object.

**Color Grounding and Score Mechanism.** We propose to use a perceptually grounded, multi-metric evaluation protocol. Instead of direct color metrics like DeltaE that penalize lighting variations, we extract RGB pixels from predicted masks and transform them to CIELAB space denoted as $\mathbf{P} = (L_i^*, a_i^*, b_i^*)_{i=1}^N$. The object may exhibit polychromatic color distribution due to geometric and lighting variations, but human observers typically abstract these variations, attributing a single representative color to an object. To capture this fundamental aspect of human vision, we adopt the dominant hue concept which is explored by (Witzel & Dewis, 2022), which identifies the representative color of an object by focusing on primary direction of chromatic variation within its color distribution. Then, we perform principal component analysis on the chromatic components ($a*$ and $b*$) of the CIELAB pixel values. It is noted by (Witzel & Dewis, 2022) that the first component $\mathbf{v}_1 = (v_{1a}, v_{1b})$ of chromaticity distribution $\mathbf{P}_{ab} = (a_i^*, b_i^*)_{i=1}^N$ represents the dominant hue. Then, chromaticity of $a_i^*, b_i^*$ is projected onto this dominant hue direction $\mathbf{v}_1$ and mean of lightness ($\overline{L}^*$) and the projected chromatic values ($\overline{a_{\text{proj}}^*}, \overline{b_{\text{proj}}^*}$) are computed to obtain the dominant color.

Now, we have the dominant color of the object and ground truth color from ISCC-NBS or CSS3/X11 color sets. However, a key challenge arises: can a single nominal color label—such as "pink" from ISCC–NBS Level 1—adequately represent the full perceptual gamut of that color category? In practice, a dominant color may correspond to a slightly different but perceptually indistinguishable shade. To account for this variability and avoid penalizing perceptually plausible matches, we construct a candidate set for each ground-truth color by including the nominal color along with its *k* perceptually nearest neighbors in the same color-naming system.

We compute three complementary metrics: (i) Delta Chroma — the Euclidean distance in $a^*, b^*$ chromaticity plane, (ii) CIEDE2000 — distribution level distance between in $L^*$, $a^*$, $b^*$ space, and (iii) MAE (Hue) — an angular difference in hue, computed in polar coordinates with chroma-based reliability gating. For each metric, we compute the minimum perceptual distance between the predicted dominant color and the candidate set. This distance is compared against the metric-specific JND threshold (typically 5), with binary scores assigned based on whether the distance falls below the threshold. An overall "Correct" assessment requires all metrics to pass.

# 4 BENCHMARK

Most existing benchmarks assess color fidelity in text-to-image generation using VQA-based approaches, as summarized in Table 1. However, these methods often rely on VLLMs that lack direct grounding in pixel-level color information, making them susceptible to hallucination, linguistic bias, and imprecise color perception. To rigorously evaluate this limitation, we constructed a controlled diagnostic set of 2464 synthetic images rendered in Blender using CSS3/X11 and ISCC–NBS L2 colors. We evaluated seven state-of-the-art VLLMs on three tasks: (i) open-ended color name/hex code prediction, (ii) multiple-choice RGB selection, and (iii) binary color verification.

| Model | Resolution | Type | Color Name Accuracy | Color-Object Association | Multi-Object Color Composition | Implicit Color Association | Numerical Color Understanding | Avg. |
|---|---|---|---|---|---|---|---|---|
| Flux | 1024 | DM | 33.70 | 18.99 | 10.49 | 22.49 | 9.14 | 18.96 |
| Sana | 1024 | DM | 49.85 | 18.10 | 7.06 | 15.18 | 15.80 | 21.20 |
| SD 3.5 | 1024 | DM | 49.83 | 20.53 | 11.43 | 17.81 | 9.41 | 21.80 |
| Pixart Alpha | 1024 | DM | 49.61 | 13.48 | 1.73 | 9.47 | 6.36 | 16.13 |
| SD 3 | 1024 | DM | 45.97 | 22.45 | 9.84 | 13.17 | 7.45 | 19.78 |
| Pixart Sigma | 1024 | DM | 47.36 | 16.75 | 3.05 | 11.49 | 6.47 | 17.02 |
| Janus Pro | 384 | AR | 29.55 | 16.33 | 8.25 | 17.88 | 3.66 | 15.13 |
| OmniGen2 | 512 | AR | 42.47 | 23.71 | 9.91 | 18.51 | 17.49 | 22.42 |
| Blip3o | 1024 | MM | 40.59 | 15.59 | 5.21 | 21.35 | 28.31 | 22.21 |

Table 4: Overall performance of T2I models on GenColorBench. *The scores are averaged over ISCC-NBS L2, L3, and CSS3/X11 colors.* ▮ ▮ ▮ incidate best, second-best, and third-best.

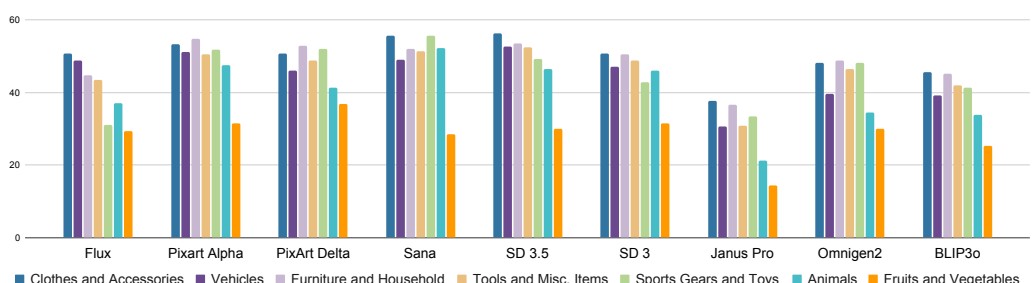

Figure 2: Performance of T2I model on category-wise color accuracy. *The scores are averaged over the Level 2 and Level 3 ISCC-NBS colors, and CSS3/X11 colors based object focused prompts.*

As shown in Table 2, the best-performing VLLM (Qwen2-VL) achieves only 49.01% accuracy on L2 binary task and 24.73% on CSS MCQ task, with open-ended performance remaining critically low (below 12.17%). These results confirm that current VLLMs struggle to reliably distinguish fine-grained colors, even under ideal conditions with single-object scenes. In contrast, our proposed method achieves 96.46% accuracy on L2 and 92.00% on CSS3/X11 colors (see appendix for details).

## 4.1 EXPERIMENT SETUP

**Models.** We focus on a broad range of the recent T2I models. This includes Flux.1 (Labs, 2024); Stable Diffusion 3.5 (Stability AI, 2024) and Stable Diffusion 3 (Stability AI, 2025) from the stability AI; PixArt-$\alpha$ (Chen et al., 2023) and PixArt-$\sigma$ (Chen et al., 2024) from the PixArt family; autoregressive models such as Janus Pro (Wu et al., 2025a) and OmniGen2 (Wu et al., 2025b); multimodal model BLIP3o (Chen et al., 2025a); and Sana (Xie et al., 2024a)—an optimized model for semantic and visual grounding. These models represent diverse architectures, ranging from diffusion-based pipelines to autoregressive and hybrid approaches. Further details are provided in the Appendix.

**Image Generation.** The evaluation is performed on a set of 44,464 prompts spanning all the five tasks described in Table 3. Following the practice in existing benchmarks, we generate 4 images per prompt, and compute the average score across all the generated images. For each model, the hyper-parameters including sampling step, and image resolution are set to default to ensure fairness in comparison. Image generation is performed using Nvidia A40 GPUs.

## 4.2 OVERALL PERFORMANCE

We evaluate the performance of various T2I models on five color generation tasks using GenColor-Bench, with results summarized in Table 4. For each task, scores are averaged across color prompts derived from Levels 2 and 3 of the ISCC-NBS system and CSS/X11 color names. Despite architectural diversity — including diffusion models (DM), autoregressive models (AR), and multimodal architectures (MM) — all models exhibit a consistent trend: performance degrades as task complexity increases. OmniGen2 (Wu et al., 2025b) achieves the highest average score (22.42), followed closely by BLIP3o (22.21) and Stable Diffusion 3.5 (21.80). Notably, OmniGen2 operates at a lower resolution (512×512) compared to SD 3.5 and BLIP3o (both 1024×1024), suggesting its superior performance is not merely resolution-dependent but may reflect stronger color semantics modeling.

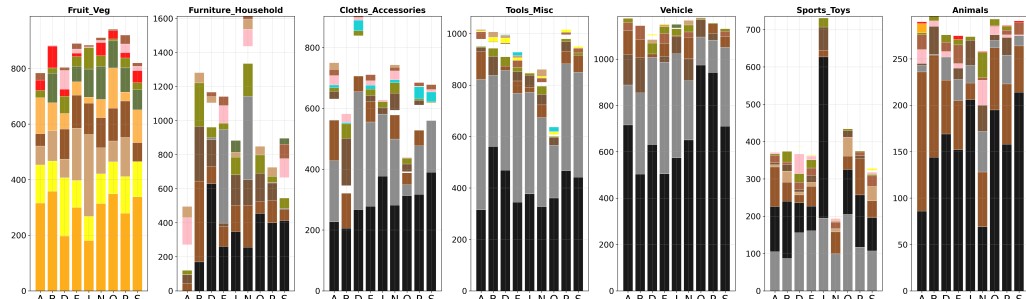

Figure 3: Distribution of estimated dominant colors (Top-10) across 10,000 generated images for each T2I models, revealing inherent color biases in vanilla baseline models. Models include: A = PixArt Alpha, B = BLIP3o, F = Flux, J = Janus-Pro, N = Sana, O = OmniGen2, P = PixArt Sigma, S = Stable Diffusion 3, and D = Stable Diffusion 3.5. *Interestingly, all the models are significantly biased towards black, gray, and brown across all the categories except fruits and vegetables.*

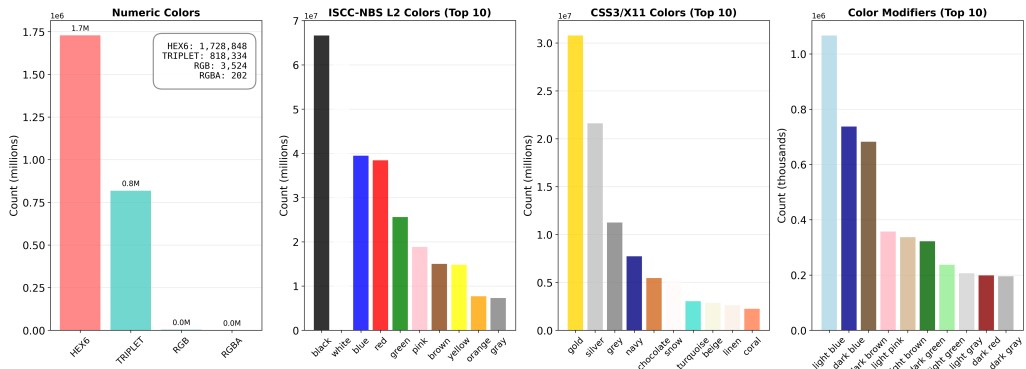

Figure 4: Color representation in LAION-2B text prompts, analyzed across four semantic categories: (i) Numeric Colors, (ii) ISCC-NBS L2 colors, (iii) CSS3/X11 named colors, and (iv) Color Modifiers. *The data reveals the dominant representation of ISCC-NBS L2 colors and their modifiers. Whereas, the numeric colors are significantly under-represented as compared to the named colors.*

On task-specific metrics, Stable Diffusion 3.5 (49.83) and Sana (49.85) lead in Color Name Accuracy, indicating strong grounding of color names, though even top performers remain below 50%, revealing persistent difficulty with fine-grained or ambiguous color terms. In contrast, performance plummets in the Color-Object Association task, where only OmniGen2 exceeds 23% (23.71), underscoring widespread failure in assigning colors to specific objects without leakage or misattribution. The Multi-Object Color Composition task reveals a sharp drop in performance across all models — with scores generally below 12 — highlighting severe limitations in spatially disentangling and assigning distinct colors to multiple objects simultaneously. Similarly, in the Implicit Color Association task, models struggle to infer color relationships embedded in texture, context, or scene semantics, with scores rarely exceeding 23%. Finally, the Numerical Color Understanding task proves most challenging, with most models scoring under 10%. Interestingly, BLIP3o significantly outperforms others here (28.31), suggesting its multimodal architecture may better encode or reason about explicit numeric color representations (e.g., RGB/hex values), which are typically learned implicitly in conventional T2I pipelines. These results collectively demonstrate that while modern T2I models can approximate basic color naming, they remain fundamentally limited in their ability to precisely control, associate, or numerically interpret color within complex visual compositions.

### 4.3 CATEGORY-LEVEL ANALYSIS

We evaluate how T2I models ground color names across seven semantic object categories as shown in Figure 2. A clear pattern emerges: models consistently achieve higher accuracy on categories such as *Clothes and Accessories*, *Vehicles*, and *Furniture and Household*, where color is often stylistic or decorative rather than semantically bound to identity. In contrast, performance drops sharply for *Animals* and *Fruits and Vegetables*, where color is biologically intrinsic (e.g., yellow banana) and

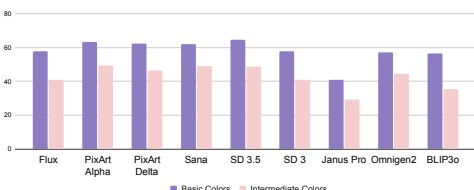 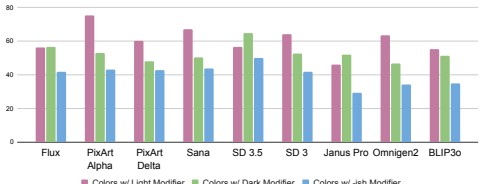

Figure 5: (Left) Comparison b/w basic and intermediate colors. These models better understand basic colors, while accuracy drops by 8–20% on intermediate colors. (Right) Comparison of color modifiers. These models understand light color modifiers better, while -ish modifiers remain worst.

requires precise disentanglement of object identity from color attribute. This disparity reflects a deep-seated training data biases. As revealed in Figure 3, all models exhibit strong chromatic bias toward *black*, *gray*, and *brown* across nearly all categories, mirroring the dominant color distribution observed in LAION-2B text prompts in Figure 4. Notably, neutral tones are overrepresented in training corpora, particularly in Vehicles and Furniture category, which explains models' relative success there. Conversely, vibrant or biologically specific colors such as reds, yellows are underrepresented in both training prompts and generated outputs, especially for Animals, and Fruits and Vegetables.

This alignment between model output bias and dataset statistics suggests that current T2I systems largely rely on statistical co-occurrence patterns rather than compositional reasoning about color semantics. For instance, the persistent rendering of bananas as "yellow" stems not from learning biological color norms, but from memorizing frequent associations in the training corpus — a phenomenon consistent with prior findings on human color-concept associations (Rathore et al., 2019). OmniGen2 and Stable Diffusion 3.5 show better cross-category generalization, while Janus Pro and BLIP3o exhibit the weakest performance, particularly struggling with color control in biologically constrained categories. This highlights that compositional color control remains challenging when decoupling color from object identity.

### 4.4 BASIC AND INTERMEDIATE COLOR UNDERSTANDING

We evaluate T2I models on basic and intermediate color understanding. To achieve this, we categorize the Red, Orange, Brown, Yellow, Olive, Yellow, Green, Blue, Purple, White, Gray, and Black as basic colors —similar to conventional color naming approaches (Berlin & Kay, 1991) where colors are described with a single word. We then group all the rest of Level 2 colors as intermediate colors. We measure the accuracy of these categories using the color naming accuracy task and illustrate the results in Figure 5(Left). These results indicate that all models perform well on basic colors, but consistently struggle with intermediate color grounding, which proves to be a more difficult task. Interestingly, there is not a large difference in the order of the models with both sets of colors, being Sana, Stable Diffusion 3.5, and PixArt-Alpha the ones obtaining best results for both type of colors.

### 4.5 MODIFIER-BASED COMPOSITIONALITY

We also analyse the understanding of color modifiers (i.e., dark, light, -ish) in T2I models. These modifiers are commonly used in natural languages to define different variants of the basic colors, e.g. light blue, dark blue, and greenish blue. Therefore, we group the ISCC-NBS Level 3 colors based on these three modifiers and study the color name accuracy task for each group. The results in terms of accuracy are shown in Figure 5(Right) which demonstrate that these models perform better with light modified colors, as compared to the dark modified colors. On the other hand, -ish modified colors remain a hard task for all the models with the performance often below than 35%, highlighting that these models struggle with gradient color semantics described in natural language.

## 5 CONCLUSIONS

We introduce GenColorBench, the first comprehensive benchmark for assessing color generation accuracy of T2I models. Our analysis of state-of-the-art models and reveals significant limitations in their ability to adhere to precise color specifications, highlighting the need for improved color controllability. GenColorBench's focus on both categorical color names and numerical values (RGB, hex) fills a key void in existing evaluation frameworks, providing a robust tool for measuring progress in this essential dimension. By establishing baseline metrics and identifying failure modes, this work lays groundwork for advancing T2I models' fidelity to color prompts.

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
