# APPENDIX:

# GENCOLORBENCH: A COLOR EVALUATION BENCHMARK FOR TEXT-TO-IMAGE GENERATION MODELS

## A APPENDIX: STATEMENTS

**Limitations/future work** Our benchmark is grounded in English-based color naming systems (ISCC-NBS and CSS/X11), which may not fully capture cross-linguistic variations in color conceptualization (Lindner et al., 2012),The choice of English is motivated by its role as a *lingua franca* in both large-scale dataset curation and the development of foundational generative models, most of which are trained predominantly on English-aligned web data. Despite this, we believe GenColorBench provides a crucial first step toward comprehensive color evaluation frameworks and establishes essential baseline metrics that can guide future research in improving color controllability in generative models.

**Broader Impacts.** GenColorBench enhances the flexible stylization capability in text-to-image synthesis by disentangling the color and texture elements. However, it also carries potential negative implications. It could be used to generate false or misleading images, thereby spreading misinformation. If is applied to generate images of public figures, it poses a risk of infringing on personal privacy. Additionally, the automatically generated images may also touch upon copyright and intellectual property issues.

**Ethical Statement.** We acknowledge the potential ethical implications of deploying generative models, including issues related to privacy, data misuse, and the propagation of biases. All models used in this paper are publicly available. We will release the modified codes to reproduce the results of this paper. We also want to point out the potential role of customization approaches in the generation of fake news, and we encourage and support responsible usage.

**Reproducibility Statement.** To facilitate reproducibility, we will make the entire source code and scripts needed to replicate all results presented in this paper available after the peer review period. We will release the code for the novel color metric we have introduced. We conducted all experiments using publicly accessible datasets. Elaborate details of all experiments have been provided in the Appendices.

**LLM usage statement.** We used a large language model solely to aid in polishing the writing and improving the clarity of the manuscript. The model was not involved in ideation, data analysis, or deriving any of the scientific contributions presented in this work.

## B DETAILS OF COLOR TAXONOMY.

To ensure a standardized evaluation of color fidelity in text-to-image generation, we ground our analysis in two widely recognized color naming systems: the Inter-Society Color Council – National Bureau of Standards (ISCC–NBS) system and the CSS3/X11 colors. These color systems are well-established and offer both perceptually meaningful color categories and precise numerical representations, facilitating human-aligned assessments.

The ISCC–NBS system organizes colors hierarchically, making it suitable for both coarse and fine-grained color evaluation. Specifically, we use Level 2 of the ISCC–NBS color system, which comprises 29 basic color categories. These represent broad, commonly recognized color terms grounded

| ID | Color Name | R | G | B | Color |
|----|------------|-----|-----|-----|-------|
| 1 | Pink | 230 | 134 | 151 | |
| 2 | Red | 185 | 40 | 66 | |
| 3 | Yellowish pink | 234 | 154 | 144 | |
| 4 | Reddish orange | 215 | 71 | 42 | |
| 5 | Reddish brown | 122 | 44 | 38 | |
| 6 | Orange | 220 | 125 | 52 | |
| 7 | Brown | 127 | 72 | 41 | |
| 8 | Orange yellow | 227 | 160 | 69 | |
| 9 | Yellowish brown | 151 | 107 | 57 | |
| 10 | Yellow | 217 | 180 | 81 | |
| 11 | Olive brown | 127 | 97 | 41 | |
| 12 | Greenish yellow | 208 | 196 | 69 | |
| 13 | Olive | 114 | 103 | 44 | |
| 14 | Yellow green | 160 | 194 | 69 | |
| 15 | Olive green | 62 | 80 | 31 | |
| 16 | Yellowish green | 74 | 195 | 77 | |
| 17 | Green | 79 | 191 | 154 | |
| 18 | Bluish green | 67 | 189 | 184 | |
| 19 | Greenish blue | 62 | 166 | 198 | |
| 20 | Blue | 59 | 116 | 192 | |
| 21 | Purplish blue | 79 | 71 | 198 | |
| 22 | Violet | 120 | 66 | 197 | |
| 23 | Purple | 172 | 74 | 195 | |
| 24 | Reddish purple | 187 | 48 | 164 | |
| 25 | Purplish pink | 229 | 137 | 191 | |
| 26 | Purplish red | 186 | 43 | 119 | |
| 27 | White | 231 | 225 | 233 | |
| 28 | Gray | 147 | 142 | 147 | |
| 29 | Black | 43 | 41 | 43 | |

Table 1: ISCC-NBS L2 Color Names and RGB Values

in perceptual uniformity. Moreover, we also use Level 3 colors, a finer-grained extension consisting of 267 distinct color names that serve as subcategories of the Level 2 set. This color set provides variation (e.g., moderate red, deep yellowish green, light bluish purple) and allow us to examine the generative models' sensitivity to subtle differences in hue, saturation, and brightness.

In addition, we incorporate the CSS3/X11 color specification, a standard widely used in web development. This set consists of 147 named colors (e.g., dodgerblue, crimson, darkslategray), each with predefined RGB values and hex color codes. These color names are familiar to a broad audience and offer an alternative taxonomy that complements the ISCC–NBS system with prevalent color terms.

We provide detailed summaries of all the color sets used in our evaluation. The ISCC–NBS Level 2 colors are listed in Table 1. ISCC–NBS Level 3 colors are listed in Table 2. While, the CSS3/X11 color set along with their RGB and hex codes are listed in Table 3.

## C  OBJECT CATEGORIZATION

To ensure a thorough evaluation of color generation in text-to-image (T2I) generation, we curate a set of 108 diverse objects that span a wide range of visual and semantic categories. This object set allows us to test how well generative models handle color prompts across different shapes, materials, and contexts. We draw these objects from two well-established datasets: COCO(Lin et al., 2014) and ImageNet(Deng et al., 2009), both of which offer a large pool of visually distinct and commonly recognized objects. Each object is selected with careful consideration of three criteria: (1) recognizability in T2I generation, ensuring that current models can reliably render the object given a prompt (e.g., "a blue chair"), (2) plausible color variability, allowing the object to appear in a wide range of

| ID | Color Name | R | G | B | ID | Color Name | R | G | B | ID | Color Name | R | G | B |
|---|---|---|---|---|---|---|---|---|---|---|---|---|---|---|
| 1 | Vivid pink | 253 | 121 | 146 | 90 | Grayish yellow | 200 | 177 | 139 | 179 | Deep blue | 17 | 48 | 116 |
| 2 | Strong pink | 244 | 143 | 160 | 91 | Dark grayish yellow | 169 | 144 | 102 | 180 | Very light blue | 153 | 198 | 249 |
| 3 | Deep pink | 230 | 105 | 128 | 92 | Yellowish white | 238 | 223 | 218 | 181 | Light blue | 115 | 164 | 220 |
| 4 | Light pink | 248 | 195 | 206 | 93 | Yellowish gray | 198 | 185 | 177 | 182 | Moderate blue | 52 | 104 | 158 |
| 5 | Moderate pink | 226 | 163 | 174 | 94 | Light olive brown | 153 | 119 | 54 | 183 | Dark blue | 23 | 52 | 89 |
| 6 | Dark pink | 197 | 128 | 138 | 95 | Moderate olive brown | 112 | 84 | 32 | 184 | Very pale blue | 194 | 210 | 236 |
| 7 | Pale pink | 239 | 209 | 220 | 96 | Dark olive brown | 63 | 44 | 16 | 185 | Pale blue | 145 | 162 | 187 |
| 8 | Grayish pink | 203 | 173 | 183 | 97 | Vivid greenish yellow | 235 | 221 | 33 | 186 | Grayish blue | 84 | 104 | 127 |
| 9 | Pinkish white | 239 | 221 | 229 | 98 | Brilliant greenish yellow | 233 | 220 | 85 | 187 | Dark grayish blue | 50 | 63 | 78 |
| 10 | Pinkish gray | 199 | 182 | 189 | 99 | Strong greenish yellow | 196 | 184 | 39 | 188 | Blackish blue | 30 | 37 | 49 |
| 11 | Vivid red | 213 | 28 | 60 | 100 | Deep greenish yellow | 162 | 152 | 18 | 189 | Bluish white | 225 | 225 | 241 |
| 12 | Strong red | 191 | 52 | 75 | 101 | Light greenish yellow | 233 | 221 | 138 | 190 | Light bluish gray | 183 | 184 | 198 |
| 13 | Deep red | 135 | 18 | 45 | 102 | Moderate greenish yellow | 192 | 181 | 94 | 191 | Bluish gray | 131 | 135 | 147 |
| 14 | Very deep red | 92 | 6 | 37 | 103 | Dark greenish yellow | 158 | 149 | 60 | 192 | Dark bluish gray | 80 | 84 | 95 |
| 15 | Moderate red | 177 | 73 | 85 | 104 | Pale greenish yellow | 230 | 220 | 171 | 193 | Bluish black | 36 | 39 | 46 |
| 16 | Dark red | 116 | 36 | 52 | 105 | Grayish greenish yellow | 190 | 181 | 132 | 194 | Vivid purplish blue | 68 | 54 | 209 |
| 17 | Very dark red | 72 | 17 | 39 | 106 | Light olive | 139 | 125 | 46 | 195 | Brilliant purplish blue | 128 | 136 | 226 |
| 18 | Light grayish red | 180 | 136 | 141 | 107 | Moderate olive | 100 | 89 | 26 | 196 | Strong purplish blue | 83 | 89 | 181 |
| 19 | Grayish red | 152 | 93 | 98 | 108 | Dark olive | 53 | 46 | 10 | 197 | Deep purplish blue | 42 | 40 | 111 |
| 20 | Dark grayish red | 83 | 56 | 62 | 109 | Light grayish olive | 142 | 133 | 111 | 198 | Very light purplish blue | 183 | 192 | 248 |
| 21 | Blackish red | 51 | 33 | 39 | 110 | Grayish olive | 93 | 85 | 63 | 199 | Light purplish blue | 137 | 145 | 203 |
| 22 | Reddish gray | 146 | 129 | 134 | 111 | Dark grayish olive | 53 | 48 | 28 | 200 | Moderate purplish blue | 77 | 78 | 135 |
| 23 | Dark reddish gray | 93 | 78 | 83 | 112 | Light olive gray | 143 | 135 | 127 | 201 | Dark purplish blue | 34 | 34 | 72 |
| 24 | Reddish black | 48 | 38 | 43 | 113 | Olive gray | 88 | 81 | 74 | 202 | Very pale purplish blue | 197 | 201 | 240 |
| 25 | Vivid yellowish pink | 253 | 126 | 93 | 114 | Olive black | 35 | 33 | 28 | 203 | Pale purplish blue | 142 | 146 | 183 |
| 26 | Strong yellowish pink | 245 | 144 | 128 | 115 | Vivid yellow green | 167 | 220 | 38 | 204 | Grayish purplish blue | 73 | 77 | 113 |
| 27 | Deep yellowish pink | 239 | 99 | 102 | 116 | Brilliant yellow green | 195 | 223 | 105 | 205 | Vivid violet | 121 | 49 | 211 |
| 28 | Light yellowish pink | 248 | 196 | 182 | 117 | Strong yellow green | 130 | 161 | 43 | 206 | Brilliant violet | 152 | 127 | 220 |
| 29 | Moderate yellowish pink | 226 | 166 | 152 | 118 | Deep yellow green | 72 | 108 | 14 | 207 | Strong violet | 97 | 65 | 156 |
| 30 | Dark yellowish pink | 201 | 128 | 126 | 119 | Light yellow green | 206 | 219 | 159 | 208 | Deep violet | 60 | 22 | 104 |
| 31 | Pale yellowish pink | 241 | 211 | 209 | 120 | Moderate yellow green | 139 | 154 | 95 | 209 | Very light violet | 201 | 186 | 248 |
| 32 | Grayish yellowish pink | 203 | 172 | 172 | 121 | Pale yellow green | 215 | 215 | 193 | 210 | Light violet | 155 | 140 | 202 |
| 33 | Brownish pink | 203 | 175 | 167 | 122 | Grayish yellow green | 151 | 154 | 133 | 211 | Moderate violet | 92 | 73 | 133 |
| 34 | Vivid reddish orange | 232 | 59 | 27 | 123 | Strong olive green | 44 | 85 | 6 | 212 | Dark violet | 52 | 37 | 77 |
| 35 | Strong reddish orange | 219 | 93 | 59 | 125 | Moderate olive green | 73 | 91 | 34 | 213 | Very pale violet | 208 | 198 | 239 |
| 36 | Deep reddish orange | 175 | 51 | 24 | 126 | Dark olive green | 32 | 52 | 11 | 214 | Pale violet | 154 | 144 | 181 |
| 37 | Moderate reddish orange | 205 | 105 | 82 | 127 | Grayish olive green | 84 | 89 | 71 | 215 | Grayish violet | 88 | 78 | 114 |
| 38 | Dark reddish orange | 162 | 64 | 43 | 128 | Dark grayish olive green | 47 | 51 | 38 | 216 | Vivid purple | 185 | 53 | 213 |
| 39 | Grayish reddish orange | 185 | 117 | 101 | 129 | Vivid yellowish green | 63 | 215 | 64 | 217 | Brilliant purple | 206 | 140 | 227 |
| 40 | Strong reddish brown | 139 | 28 | 14 | 130 | Brilliant yellowish green | 135 | 217 | 137 | 218 | Strong purple | 147 | 82 | 168 |
| 41 | Deep reddish brown | 97 | 15 | 18 | 131 | Strong yellowish green | 57 | 150 | 74 | 219 | Deep purple | 101 | 34 | 119 |
| 42 | Light reddish brown | 172 | 122 | 115 | 132 | Deep yellowish green | 23 | 106 | 30 | 220 | Very deep purple | 70 | 10 | 85 |
| 43 | Moderate reddish brown | 125 | 66 | 59 | 133 | Very deep yellowish green | 5 | 66 | 8 | 221 | Very light purple | 228 | 185 | 243 |
| 44 | Dark reddish brown | 70 | 29 | 30 | 134 | Very light yellowish green | 197 | 237 | 196 | 222 | Light purple | 188 | 147 | 204 |
| 45 | Light grayish reddish brown | 158 | 127 | 122 | 135 | Light yellowish green | 156 | 198 | 156 | 223 | Moderate purple | 135 | 94 | 150 |
| 46 | Grayish reddish brown | 108 | 77 | 75 | 136 | Moderate yellowish green | 102 | 144 | 105 | 224 | Dark purple | 86 | 55 | 98 |
| 47 | Dark grayish reddish brown | 67 | 41 | 42 | 137 | Dark yellowish green | 47 | 93 | 58 | 225 | Very dark purple | 55 | 27 | 65 |
| 48 | Vivid orange | 247 | 118 | 11 | 138 | Very dark yellowish green | 16 | 54 | 26 | 226 | Very pale purple | 224 | 203 | 235 |
| 50 | Strong orange | 234 | 129 | 39 | 139 | Vivid green | 35 | 234 | 165 | 227 | Pale purple | 173 | 151 | 179 |
| 51 | Deep orange | 194 | 96 | 18 | 140 | Brilliant green | 73 | 208 | 163 | 228 | Grayish purple | 123 | 102 | 126 |
| 52 | Light orange | 251 | 175 | 130 | 141 | Strong green | 21 | 138 | 102 | 229 | Dark grayish purple | 81 | 63 | 81 |
| 53 | Moderate orange | 222 | 141 | 92 | 143 | Very light green | 166 | 226 | 202 | 230 | Blackish purple | 47 | 34 | 49 |
| 54 | Brownish orange | 178 | 102 | 51 | 144 | Light green | 111 | 172 | 149 | 231 | Purplish white | 235 | 223 | 239 |
| 55 | Strong brown | 138 | 68 | 22 | 145 | Moderate green | 51 | 119 | 98 | 232 | Light purplish gray | 195 | 183 | 198 |
| 56 | Deep brown | 87 | 26 | 7 | 146 | Dark green | 22 | 78 | 61 | 233 | Purplish gray | 143 | 132 | 144 |
| 57 | Light brown | 173 | 124 | 99 | 147 | Very dark green | 12 | 46 | 36 | 234 | Dark purplish gray | 92 | 82 | 94 |
| 58 | Moderate brown | 114 | 74 | 56 | 148 | Very pale green | 199 | 217 | 214 | 235 | Purplish black | 43 | 38 | 48 |
| 59 | Dark brown | 68 | 33 | 18 | 149 | Pale green | 148 | 166 | 163 | 236 | Vivid reddish purple | 212 | 41 | 185 |
| 60 | Light grayish brown | 153 | 127 | 117 | 150 | Grayish green | 97 | 113 | 110 | 237 | Strong reddish purple | 167 | 73 | 148 |
| 61 | Grayish brown | 103 | 79 | 72 | 151 | Dark grayish green | 57 | 71 | 70 | 238 | Deep reddish purple | 118 | 26 | 106 |
| 62 | Dark grayish brown | 62 | 44 | 40 | 152 | Blackish green | 31 | 42 | 42 | 239 | Very deep reddish purple | 79 | 9 | 74 |
| 63 | Light brownish gray | 146 | 130 | 129 | 153 | Greenish white | 224 | 226 | 229 | 240 | Light reddish purple | 189 | 128 | 174 |
| 64 | Brownish gray | 96 | 82 | 81 | 154 | Light greenish gray | 186 | 190 | 193 | 241 | Moderate reddish purple | 150 | 88 | 136 |
| 65 | Brownish black | 43 | 33 | 30 | 155 | Greenish gray | 132 | 136 | 136 | 242 | Dark reddish purple | 95 | 52 | 88 |
| 67 | Brilliant orange yellow | 255 | 190 | 80 | 156 | Dark greenish gray | 84 | 88 | 88 | 243 | Very dark reddish purple | 63 | 24 | 60 |
| 68 | Strong orange yellow | 240 | 161 | 33 | 157 | Greenish black | 33 | 38 | 38 | 244 | Pale reddish purple | 173 | 137 | 165 |
| 69 | Deep orange yellow | 208 | 133 | 17 | 158 | Vivid bluish green | 19 | 252 | 213 | 245 | Grayish reddish purple | 134 | 98 | 126 |
| 70 | Light orange yellow | 252 | 194 | 124 | 159 | Brilliant bluish green | 53 | 215 | 206 | 246 | Brilliant purplish pink | 252 | 161 | 231 |
| 71 | Moderate orange yellow | 231 | 167 | 93 | 160 | Strong bluish green | 13 | 143 | 130 | 247 | Strong purplish pink | 244 | 131 | 205 |
| 72 | Dark orange yellow | 195 | 134 | 57 | 162 | Very light bluish green | 152 | 225 | 224 | 248 | Deep purplish pink | 223 | 106 | 172 |
| 73 | Pale orange yellow | 238 | 198 | 166 | 163 | Light bluish green | 95 | 171 | 171 | 249 | Light purplish pink | 245 | 178 | 219 |
| 74 | Strong yellowish brown | 158 | 103 | 29 | 164 | Moderate bluish green | 41 | 122 | 123 | 250 | Moderate purplish pink | 222 | 152 | 191 |
| 75 | Deep yellowish brown | 103 | 63 | 11 | 165 | Dark bluish green | 21 | 75 | 77 | 251 | Dark purplish pink | 198 | 125 | 157 |
| 76 | Light yellowish brown | 196 | 154 | 116 | 166 | Very dark bluish green | 10 | 45 | 46 | 252 | Pale purplish pink | 235 | 200 | 223 |
| 77 | Moderate yellowish brown | 136 | 102 | 72 | 168 | Brilliant greenish blue | 45 | 188 | 226 | 253 | Grayish purplish pink | 199 | 163 | 185 |
| 78 | Dark yellowish brown | 80 | 52 | 26 | 169 | Strong greenish blue | 19 | 133 | 175 | 254 | Vivid purplish red | 221 | 35 | 136 |
| 79 | Light grayish yellowish brown | 180 | 155 | 141 | 171 | Very light greenish blue | 148 | 214 | 239 | 255 | Strong purplish red | 184 | 55 | 115 |
| 80 | Grayish yellowish brown | 126 | 105 | 93 | 172 | Light greenish blue | 101 | 168 | 195 | 256 | Deep purplish red | 136 | 16 | 85 |
| 81 | Dark grayish yellowish brown | 77 | 61 | 51 | 173 | Moderate greenish blue | 42 | 118 | 145 | 257 | Very deep purplish red | 84 | 6 | 60 |
| 82 | Vivid yellow | 241 | 191 | 21 | 174 | Dark greenish blue | 19 | 74 | 96 | 258 | Moderate purplish red | 171 | 75 | 116 |
| 83 | Brilliant yellow | 247 | 206 | 80 | 175 | Very dark greenish blue | 11 | 44 | 59 | 259 | Dark purplish red | 110 | 41 | 76 |
| 84 | Strong yellow | 217 | 174 | 47 | 176 | Vivid blue | 27 | 92 | 215 | 260 | Very dark purplish red | 67 | 20 | 50 |
| 85 | Deep yellow | 184 | 143 | 22 | 177 | Brilliant blue | 65 | 157 | 237 | 261 | Light grayish purplish red | 178 | 135 | 155 |
| 86 | Light yellow | 244 | 210 | 132 | 178 | Strong blue | 39 | 108 | 189 | 262 | Grayish purplish red | 148 | 92 | 115 |
| 87 | Moderate yellow | 210 | 175 | 99 |  |  |  |  |  | 263 | White | 231 | 225 | 233 |
| 88 | Dark yellow | 176 | 143 | 66 |  |  |  |  |  | 264 | Light gray | 189 | 183 | 191 |
| 89 | Pale yellow | 239 | 215 | 178 |  |  |  |  |  | 265 | Medium gray | 138 | 132 | 137 |
|  |  |  |  |  |  |  |  |  |  | 266 | Dark gray | 88 | 84 | 88 |
|  |  |  |  |  |  |  |  |  |  | 267 | Black | 43 | 41 | 43 |

Table 2: List of ISCC NBS Level 3 Colors used in the GenColorBench evaluation

| ID | Color Name | {hex} | R | G | B | ID | Color Name | {hex} | R | G | B | ID | Color Name | {hex} | R | G | B |
|----|-----------|-------|---|---|---|----|-----------|-------|---|---|---|----|-----------|-------|---|---|---|
| 1 | AliceBlue | #F0F8FF | 240 | 248 | 255 | 50 | LightBlue | #ADD8E6 | 173 | 216 | 230 | 99 | PowderBlue | #B0E0E6 | 176 | 224 | 230 |
| 2 | AntiqueWhite | #FAEBD7 | 250 | 235 | 215 | 51 | LightCoral | #F08080 | 240 | 128 | 128 | 100 | Purple | #800080 | 128 | 0 | 128 |
| 3 | Aqua | #00FFFF | 0 | 255 | 255 | 52 | LightCyan | #E0FFFF | 224 | 255 | 255 | 101 | RebeccaPurple | #663399 | 102 | 51 | 153 |
| 4 | Aquamarine | #7FFFD4 | 127 | 255 | 212 | 53 | LightGoldenRodYellow | #FAFAD2 | 250 | 250 | 210 | 102 | Red | #FF0000 | 255 | 0 | 0 |
| 5 | Azure | #F0FFFF | 240 | 255 | 255 | 54 | LightGray | #D3D3D3 | 211 | 211 | 211 | 103 | RosyBrown | #BC8F8F | 188 | 143 | 143 |
| 6 | Beige | #F5F5DC | 245 | 245 | 220 | 55 | LightGrey | #D3D3D3 | 211 | 211 | 211 | 104 | RoyalBlue | #4169E1 | 65 | 105 | 225 |
| 7 | Bisque | #FFE4C4 | 255 | 228 | 196 | 56 | LightGreen | #90EE90 | 144 | 238 | 144 | 105 | SaddleBrown | #8B4513 | 139 | 69 | 19 |
| 8 | Black | #000000 | 0 | 0 | 0 | 57 | LightPink | #FFB6C1 | 255 | 182 | 193 | 106 | Salmon | #FA8072 | 250 | 128 | 114 |
| 9 | BlanchedAlmond | #FFEBCD | 255 | 235 | 205 | 58 | LightSalmon | #FFA07A | 255 | 160 | 122 | 107 | SandyBrown | #F4A460 | 244 | 164 | 96 |
| 10 | Blue | #0000FF | 0 | 0 | 255 | 59 | LightSeaGreen | #20B2AA | 32 | 178 | 170 | 108 | SeaGreen | #2E8B57 | 46 | 139 | 87 |
| 11 | BlueViolet | #8A2BE2 | 138 | 43 | 226 | 60 | LightSkyBlue | #87CEFA | 135 | 206 | 250 | 109 | SeaShell | #FFF5EE | 255 | 245 | 238 |
| 12 | Brown | #A52A2A | 165 | 42 | 42 | 61 | LightSlateGray | #778899 | 119 | 136 | 153 | 110 | Sienna | #A0522D | 160 | 82 | 45 |
| 13 | BurlyWood | #DEB887 | 222 | 184 | 135 | 62 | LightSlateGrey | #778899 | 119 | 136 | 153 | 111 | Silver | #C0C0C0 | 192 | 192 | 192 |
| 14 | CadetBlue | #5F9EA0 | 95 | 158 | 160 | 63 | LightSteelBlue | #B0C4DE | 176 | 196 | 222 | 112 | SkyBlue | #87CEEB | 135 | 206 | 235 |
| 15 | Chartreuse | #7FFF00 | 127 | 255 | 0 | 64 | LightYellow | #FFFFE0 | 255 | 255 | 224 | 113 | SlateBlue | #6A5ACD | 106 | 90 | 205 |
| 16 | Chocolate | #D2691E | 210 | 105 | 30 | 65 | Lime | #00FF00 | 0 | 255 | 0 | 114 | SlateGray | #708090 | 112 | 128 | 144 |
| 17 | Coral | #FF7F50 | 255 | 127 | 80 | 66 | LimeGreen | #32CD32 | 50 | 205 | 50 | 115 | SlateGrey | #708090 | 112 | 128 | 144 |
| 18 | CornflowerBlue | #6495ED | 100 | 149 | 237 | 67 | Linen | #FAF0E6 | 250 | 240 | 230 | 116 | Snow | #FFFAFA | 255 | 250 | 250 |
| 19 | Cornsilk | #FFF8DC | 255 | 248 | 220 | 68 | Magenta | #FF00FF | 255 | 0 | 255 | 117 | SpringGreen | #00FF7F | 0 | 255 | 127 |
| 20 | Crimson | #DC143C | 220 | 20 | 60 | 69 | Maroon | #800000 | 128 | 0 | 0 | 118 | SteelBlue | #4682B4 | 70 | 130 | 180 |
| 21 | Cyan | #00FFFF | 0 | 255 | 255 | 70 | MediumAquaMarine | #66CDAA | 102 | 205 | 170 | 119 | Tan | #D2B48C | 210 | 180 | 140 |
| 22 | DarkBlue | #00008B | 0 | 0 | 139 | 71 | MediumBlue | #0000CD | 0 | 0 | 205 | 120 | Teal | #008080 | 0 | 128 | 128 |
| 23 | DarkCyan | #008B8B | 0 | 139 | 139 | 72 | MediumOrchid | #BA55D3 | 186 | 85 | 211 | 121 | Thistle | #D8BFD8 | 216 | 191 | 216 |
| 24 | DarkGoldenRod | #B8860B | 184 | 134 | 11 | 73 | MediumPurple | #9370DB | 147 | 112 | 219 | 122 | Tomato | #FF6347 | 255 | 99 | 71 |
| 25 | DarkGray | #A9A9A9 | 169 | 169 | 169 | 74 | MediumSeaGreen | #3CB371 | 60 | 179 | 113 | 123 | Turquoise | #40E0D0 | 64 | 224 | 208 |
| 26 | DarkGrey | #A9A9A9 | 169 | 169 | 169 | 75 | MediumSlateBlue | #7B68EE | 123 | 104 | 238 | 124 | Violet | #EE82EE | 238 | 130 | 238 |
| 27 | DarkGreen | #006400 | 0 | 100 | 0 | 76 | MediumSpringGreen | #00FA9A | 0 | 250 | 154 | 125 | Wheat | #F5DEB3 | 245 | 222 | 179 |
| 28 | DarkKhaki | #BDB76B | 189 | 183 | 107 | 77 | MediumTurquoise | #48D1CC | 72 | 209 | 204 | 126 | White | #FFFFFF | 255 | 255 | 255 |
| 29 | DarkMagenta | #8B008B | 139 | 0 | 139 | 78 | MediumVioletRed | #C71585 | 199 | 21 | 133 | 127 | WhiteSmoke | #F5F5F5 | 245 | 245 | 245 |
| 30 | DarkOliveGreen | #556B2F | 85 | 107 | 47 | 79 | MidnightBlue | #191970 | 25 | 25 | 112 | 128 | Yellow | #FFFF00 | 255 | 255 | 0 |
| 31 | DarkOrange | #FF8C00 | 255 | 140 | 0 | 80 | MintCream | #F5FFFA | 245 | 255 | 250 | 129 | YellowGreen | #9ACD32 | 154 | 205 | 50 |
| 32 | DarkOrchid | #9932CC | 153 | 50 | 204 | 81 | MistyRose | #FFE4E1 | 255 | 228 | 225 | | | | | | |
| 33 | DarkRed | #8B0000 | 139 | 0 | 0 | 82 | Moccasin | #FFE4B5 | 255 | 228 | 181 | | | | | | |
| 34 | DarkSalmon | #E9967A | 233 | 150 | 122 | 83 | NavajoWhite | #FFDEAD | 255 | 222 | 173 | | | | | | |
| 35 | DarkSeaGreen | #8FBC8F | 143 | 188 | 143 | 84 | Navy | #000080 | 0 | 0 | 128 | | | | | | |
| 36 | DarkSlateBlue | #483D8B | 72 | 61 | 139 | 85 | OldLace | #FDF5E6 | 253 | 245 | 230 | | | | | | |
| 37 | DarkSlateGray | #2F4F4F | 47 | 79 | 79 | 86 | Olive | #808000 | 128 | 128 | 0 | | | | | | |
| 38 | DarkSlateGrey | #2F4F4F | 47 | 79 | 79 | 87 | OliveDrab | #6B8E23 | 107 | 142 | 35 | | | | | | |
| 39 | DarkTurquoise | #00CED1 | 0 | 206 | 209 | 88 | Orange | #FFA500 | 255 | 165 | 0 | | | | | | |
| 40 | DarkViolet | #9400D3 | 148 | 0 | 211 | 89 | OrangeRed | #FF4500 | 255 | 69 | 0 | | | | | | |
| 41 | DeepPink | #FF1493 | 255 | 20 | 147 | 90 | Orchid | #DA70D6 | 218 | 112 | 214 | | | | | | |
| 42 | DeepSkyBlue | #00BFFF | 0 | 191 | 255 | 91 | PaleGoldenRod | #EEE8AA | 238 | 232 | 170 | | | | | | |
| 43 | DimGray | #696969 | 105 | 105 | 105 | 92 | PaleGreen | #98FB98 | 152 | 251 | 152 | | | | | | |
| 44 | DimGrey | #696969 | 105 | 105 | 105 | 93 | PaleTurquoise | #AFEEEE | 175 | 238 | 238 | | | | | | |
| 45 | DodgerBlue | #1E90FF | 30 | 144 | 255 | 94 | PaleVioletRed | #DB7093 | 219 | 112 | 147 | | | | | | |
| 46 | FireBrick | #B22222 | 178 | 34 | 34 | 95 | PapayaWhip | #FFEFD5 | 255 | 239 | 213 | | | | | | |
| 47 | FloralWhite | #FFFAF0 | 255 | 250 | 240 | 96 | PeachPuff | #FFDAB9 | 255 | 218 | 185 | | | | | | |
| 48 | ForestGreen | #228B22 | 34 | 139 | 34 | 97 | Peru | #CD853F | 205 | 133 | 63 | | | | | | |
| 49 | Fuchsia | #FF00FF | 255 | 0 | 255 | 98 | Pink | #FFC0CB | 255 | 192 | 203 | | | | | | |

Table 3: List of CSS3/X11 colors used in GenColorBench evaluation.

colors (e.g., a shirt or a car), and (3) segmentation suitability, so the object can be cleanly separated from its background using segmentation models.

While conducting initial experiments, we observed that many generated images include additional visual components associated with the main object, but which are not relevant for color evaluation. For example, when prompting for a "red car," the generated image may include elements such as tires, headlights, or windows—components that differ in material and expected color from the car's painted body. Including these in the mask during evaluation would introduce noise and bias in the color measurements. To address this issue, we introduce the concept of negative labels—subcomponents or associated elements of an object that should be excluded from the evaluation mask. For each object class, we generate a list of such negative labels using GPT-4o, leveraging its broad world knowledge and language understanding to identify parts that are typically not relevant for color fidelity. These negative labels help refine the segmentation masks, allowing us to more precisely isolate the region of interest (e.g., the body of a car), and ensure a fair and focused color assessment. All the objects along with their negative labels are listed in Table 4, and Table 5.

# D PROMPT TEMPLATES

We organize prompts into four levels of difficulty, based on their semantic and compositional complexity: (1) Object-Focused Prompts—These are straightforward, object-centric prompts that mention only a single object and its target color (e.g., "a red apple" or "a green chair"). The goal here is to assess basic color understanding, including both color name fidelity and numerical color consistency with the expected RGB/hex value. These prompts are the most direct and least ambiguous prompts. (2) Contextual Prompts—the object is described within a broader scene or setting, but only a single colored object is mentioned (e.g., "a blue vase on a wooden table near a window"). These prompts evaluate the model's ability to preserve the intended color within contextual descriptions, measuring both color name accuracy and the association of the correct color to the correct object in a scene. (3) Scene Descriptive Prompts—These prompts describe scenes containing multiple objects, each associated with its own color (e.g., "a red apple next to a green pear and a yellow banana").

| Dataset | Category | Class Name | Negative Labels |
|---|---|---|---|
| COCO | Vehicle | vehicle | glass windows. front glass windows. back glass windows. windshield. black wheel. tire. metal rim. headlight. taillight. mirror. metal bumper. license plate. roof rack. antenna. grille. door handle. door frame. side mirror glass. hubcap. mudguard. windshield wiper. |
| COCO | Vehicle | bicycle | wheel. black tire. rim. seat. basket. pedal. steel handle. metal handle. metal nut. chain. brake disc. reflector. gear shift. handlebar grip. spoke. |
| COCO | Vehicle | car | window. windshield. wheel. tire. rim. headlight. taillight. mirror. bumper. license plate. roof rack. antenna. grille. door handle. door frame. side mirror glass. hubcap. mudguard. windshield wiper. exhaust pipe. |
| COCO | Vehicle | motorcycle | black wheel. tire. engine. rim. headlight. taillight. mirror. exhaust. leather seat. handlebar. chain. foot peg. black handle. metal stand. brake lever. turn signal. |
| COCO | Vehicle | airplane | window. glass windows. front glass window. back glass window. plane engine. black tire. antenna. tail fin. wing flap. landing gear. cockpit window. propeller. |
| COCO | Vehicle | bus | window. front glass windows. back glass window. windshield. black wheel. tire. rim. headlight. taillight. mirror. bumper. license plate. door handle. roof rack. antenna. side mirror glass. exhaust pipe. stop sign. |
| COCO | Vehicle | train | window. door. wheel. bogie. pantograph. headlight. buffer. coupling. antenna. rail. overhead line. windshield. wiper. light. connector. ladder. handle. logo. text. number. symbol. pipe. cable. horn. vent. exhaust. panel. joint. frame. suspension. track. gravel. pole. wire. signboard. indicator. sticker. emblem. |
| COCO | Vehicle | truck | window. windshield. wheel. tire. rim. headlight. taillight. mirror. bumper. license plate. roof rack. antenna. grille. door handle. exhaust pipe. side mirror glass. mudflap. |
| COCO | Vehicle | boat | window. hull detail. motor. propeller. antenna. railing. rope. flag. deck equipment. lifebuoy. mast. anchor. |
| COCO | Fruit and Veg | banana | sticker. wrap. stem. leaf. spot. bruise. brown spot. peel. string. |
| COCO | Fruit and Veg | apple | sticker. wrap. stem. leaf. spot. bruise. stem scar. wax coating. calyx. |
| COCO | Fruit and Veg | orange | sticker. wrap. stem. leaf. spot. bruise. peel. segment. pith. |
| COCO | Fruit and Veg | broccoli | stem. leaf. spot. bruise. floret. stalk. |
| COCO | Fruit and Veg | carrot | stem. leaf. spot. dirt. root tip. soil. |
| COCO | furniture and Household | chair | cushion. leg. screw. joint. tag. upholstery. armrest. backrest. |
| COCO | furniture and Household | couch | cushion. leg. seam. button. pillow. tag. upholstery. armrest. backrest. |
| COCO | furniture and Household | potted plant | pot. soil. label. stick. tag. drainage hole. saucer. |
| COCO | furniture and Household | sink | faucet only. drain. soap dispenser. knob. handle above sink. countertop. |
| COCO | furniture and Household | book | cover. spine. bookmark. sticker. dust jacket. pages. |
| COCO | furniture and Household | clock | glass. hands. dial. numbers. knob. frame. pendulum. |
| COCO | furniture and Household | vase | rim. neck. base. chip. pattern. glaze. |
| COCO | animals | cat | eyes. teeth. nose. mouth. tongue. paw. claw. ear. whiskers. tail. fur. collar. leash. tag. accessory. fur pattern. fur texture. |
| COCO | animals | dog | eyes. teeth. nose. mouth. tongue. paw. claw. ear. whiskers. tail. fur. collar. leash. tag. accessory. fur pattern. fur texture. |
| COCO | animals | horse | eyes. teeth. nose. mouth. tongue. hoof. claw. ear. whiskers. tail. mane. fur. saddle. reins. harness. bridle. coat pattern. |
| COCO | animals | sheep | eyes. teeth. nose. mouth. tongue. hoof. claw. ear. whiskers. tail. wool. tag. collar. fleece texture. |
| COCO | animals | cow | eyes. teeth. nose. mouth. tongue. hoof. claw. ear. whiskers. tail. fur. tag. bell. collar. coat pattern. |
| COCO | animals | elephant | eyes. teeth. nose. mouth. tongue. paw. claw. ear. whiskers. tail. tusk. mane. fur. chain. tusk cover. cloth. skin folds. |
| COCO | animals | bear | eyes. teeth. nose. mouth. tongue. paw. claw. ear. whiskers. tail. fur. collar. accessory. fur pattern. |
| COCO | animals | zebra | eyes. teeth. nose. mouth. tongue. paw. claw. ear. whiskers. tail. stripes. fur. harness. accessory. |
| COCO | animals | giraffe | eyes. teeth. nose. mouth. tongue. paw. claw. ear. whiskers. tail. fur. tag. collar. spots. |
| COCO | cloths and Accessories | tie | text. logo. monogram. tag. knot. |
| COCO | cloths and Accessories | handbag | text. logo. monogram. inner layer. tag. seam. button. zipper. strap. |
| COCO | cloths and Accessories | backpack | text. logo. monogram. inner layer. tag. seam. button. zipper. strap. buckle. pocket. |
| COCO | cloths and Accessories | suitcase | text. logo. monogram. inner layer. tag. seam. button. zipper. handle. wheel. lock. zipper pull. |
| COCO | cloths and Accessories | umbrella | text. logo. monogram. tag. seam. pipe. handle. fabric folds. |
| COCO | sports and toys | sports ball | logo. text. stitching. brand name. grip tape. valve. panel seams. |
| COCO | sports and toys | baseball bat | logo. text. grip tape. brand name. handle. barrel. |
| COCO | sports and toys | kite | string. tail. frame. handle. spars. |
| COCO | sports and toys | frisbee | logo. text. brand name. rim. |
| COCO | sports and toys | surfboard | logo. text. brand name. leash. fin. deck pad. |
| COCO | sports and toys | skis | logo. text. brand name. binding. tip. tail. |
| COCO | sports and toys | baseball glove | logo. text. stitching. brand name. webbing. |
| COCO | sports and toys | skateboard | wheel. logo. text. grip tape. brand name. trucks. deck. |
| COCO | Tools and Misc | hair drier | button. switch. cord. label. nozzle. |
| COCO | Tools and Misc | remote | button. label. logo. screen. |
| COCO | Tools and Misc | microwave | button. label. logo. handle. display panel. |
| COCO | Tools and Misc | sink | faucet. drain. knob. handle. basin. |
| COCO | Tools and Misc | toaster | button. label. logo. slot. |
| COCO | Tools and Misc | refrigerator | handle. logo. label. button. door seal. |
| COCO | Tools and Misc | oven | handle. button. label. logo. control panel. |
| COCO | Tools and Misc | knife | handle. logo. brand name. blade edge. |

Table 4: List of the objects selected from the COCO dataset.

This level tests the model's ability to distinguish and correctly apply multiple colors to multiple objects, and is useful for evaluating multi-object compositionality and the avoidance of color-object entanglement. (4) Implicit Color Association—These are the most semantically complex prompts, involving color references between objects (e.g., "a cup that is the same color as the nearby blue notebook", or "a cat lying on a rug that shares its pink color"). Here, only one object is explicitly assigned a color, and the second object's color is described relationally. These prompts assess whether models can understand and generate color consistency through indirect, reference-based language. We list all the prompt templates below.

## D.1 LIST OF OBJECT FOCUSED PROMPT TEMPLATES.

1. A {color} {object}
2. The {object} is {color}
3. A photo of a {color} {object}

| Dataset | Category | Class Name | Negative Labels |
|---|---|---|---|
| ImageNet | Vehicle | ambulance | window. windshield. wheel. tire. rim. headlight. taillight. mirror. bumper. license plate. roof rack. antenna. grille. door handle. siren. |
| ImageNet | Vehicle | beach wagon | wheel. tire. rim. seat. handlebar. pedal. chain. brake. reflector. bell. gear. |
| ImageNet | Vehicle | jeep | window. windshield. wheel. tire. rim. headlight. taillight. mirror. bumper. license plate. roof rack. antenna. grille. door handle. |
| ImageNet | Vehicle | minivan | window. windshield. wheel. tire. rim. headlight. taillight. mirror. bumper. license plate. roof rack. antenna. grille. door handle. |
| ImageNet | Vehicle | sports car | window. windshield. wheel. tire. rim. headlight. taillight. mirror. bumper. license plate. roof rack. antenna. grille. door handle. |
| ImageNet | Vehicle | tow truck | window. windshield. wheel. tire. rim. headlight. taillight. mirror. bumper. license plate. roof rack. antenna. grille. door handle. tow hook. |
| ImageNet | Vehicle | ferry | window. hull detail. motor. propeller. antenna. railing. rope. flag. deck equipment. lifeboat. |
| ImageNet | Vehicle | taxi | window. windshield. wheel. tire. rim. headlight. taillight. mirror. bumper. license plate. roof rack. antenna. grille. door handle. taxi sign. |
| ImageNet | Fruit and Veg | lemon | sticker. wrap. stem. leaf. spot. bruise. peel texture. |
| ImageNet | Fruit and Veg | mango | sticker. wrap. stem. leaf. spot. bruise. peel texture. |
| ImageNet | Fruit and Veg | papaya | sticker. wrap. stem. leaf. spot. bruise. seeds. peel texture. |
| ImageNet | Fruit and Veg | guava | sticker. wrap. stem. leaf. spot. bruise. seeds. peel texture. |
| ImageNet | Fruit and Veg | strawberry | sticker. wrap. stem. leaf. spot. bruise. seeds. calyx. |
| ImageNet | furniture and Household | teapot | lid. handle. spout. base. knob. |
| ImageNet | furniture and Household | table | leg. joint. screw. tabletop. |
| ImageNet | furniture and Household | desk | leg. joint. screw. drawer. |
| ImageNet | furniture and Household | bookcase | shelf. joint. screw. back panel. |
| ImageNet | furniture and Household | wardrobe | handle. knob. hinge. door. |
| ImageNet | furniture and Household | mug | handle. rim. base. |
| ImageNet | furniture and Household | candle | wick. flame. holder. wax. |
| ImageNet | animals | tiger | eyes. teeth. nose. mouth. tongue. paw. claw. ear. whiskers. tail. stripes. fur. mane. |
| ImageNet | animals | parrot | beak. claw. wingtip. feather. cage. perch. tag. |
| ImageNet | animals | duck | beak. claw. wingtip. feather. tag. accessory. |
| ImageNet | animals | crocodile | eyes. teeth. scales. tail. claw. |
| ImageNet | animals | shark | eyes. teeth. fin. tail. gills. |
| ImageNet | animals | lobster | claw band. eyes. antenna. legs. shell. |
| ImageNet | animals | goldfish | eyes. fins. tail. bowl. accessory. |
| ImageNet | animals | turtle | eyes. shell. legs. tail. scales. |
| ImageNet | animals | owl | eyes. beak. wingtip. feather. talons. |
| ImageNet | cloths and Accessories | T-shirt | text. logo. monogram. inner layer. tag. seam. button. collar. cuff. pocket. |
| ImageNet | cloths and Accessories | sweatshirt | text. logo. monogram. inner layer. tag. seam. button. collar. cuff. pocket. |
| ImageNet | cloths and Accessories | suit | text. logo. monogram. inner layer. tag. seam. button. collar. cuff. pocket. |
| ImageNet | cloths and Accessories | jacket | text. logo. monogram. inner layer. tag. seam. button. collar. cuff. pocket. |
| ImageNet | cloths and Accessories | coat | text. logo. monogram. inner layer. tag. seam. button. collar. cuff. pocket. |
| ImageNet | cloths and Accessories | jeans | text. logo. monogram. inner layer. tag. seam. button. pocket. |
| ImageNet | cloths and Accessories | pants | text. logo. monogram. inner layer. tag. seam. button. pocket. |
| ImageNet | cloths and Accessories | shorts | text. logo. monogram. inner layer. tag. seam. button. pocket. |
| ImageNet | cloths and Accessories | hat | text. logo. monogram. inner layer. tag. seam. button. |
| ImageNet | sports and toys | football helmet | logo. text. stitching. brand name. padding. strap. face guard. |
| ImageNet | sports and toys | golf ball | logo. text. stitching. brand name. dimples. |
| ImageNet | sports and toys | boxing glove | logo. text. stitching. brand name. strap. |
| ImageNet | sports and toys | teddy bear | eyes. nose. mouth. paw. claw. ear. tongue. collar. tag. accessory. fur. stitching. |
| ImageNet | sports and toys | snowboard | logo. text. brand name. leash. binding. fin. |
| ImageNet | sports and toys | balloon | string. knot. valve. |
| ImageNet | sports and toys | doll | string. hair. eyes. mouth. nose. limbs. dress. accessory. tag. |
| ImageNet | sports and toys | toy poodle | eyes. nose. mouth. paw. claw. ear. tongue. collar. tag. accessory. fur. stitching. |
| ImageNet | sports and toys | toy terrier | eyes. nose. mouth. paw. claw. ear. tongue. collar. tag. accessory. fur. stitching. |
| ImageNet | Tools and Misc | sponge | label. tag. pores. |
| ImageNet | Tools and Misc | cutting board | knife marks. logo. text. |
| ImageNet | Tools and Misc | computer mouse | button. logo. cable. scroll wheel. |
| ImageNet | Tools and Misc | hair dryer | button. switch. cord. label. nozzle. |
| ImageNet | Tools and Misc | iron | button. switch. cord. label. soleplate. |
| ImageNet | Tools and Misc | fan | button. switch. cord. label. blades. |
| ImageNet | Tools and Misc | hammer | handle. brand name. logo. claw. |
| ImageNet | Tools and Misc | wrench | handle. brand name. logo. jaw. |
| ImageNet | Tools and Misc | saw | handle. brand name. logo. blade. |
| ImageNet | Tools and Misc | ruler | markings. text. logo. edges. |

Table 5: List of the objects selected from the ImageNet dataset.

4. A {object} that is entirely {color}

5. An image of a {color} {object}

6. A {color} colored {object}

7. A single {color} {object}

8. A {object}, and it's {color}

9. A {object} in a {color} color

10. A {object} rendered in {color} color

11. A {object} with a {color} color

12. A realistic {object} in {color}

13. An image of a {object} in hex color {hex}

14. A {object} in color {hex}

15. A {object} with hex color {hex}

16. A close-up of a {object} in the color {hex}

17. A {object} rendered in {hex} color

18. A photo of a {object} in the color {hex}

19. A {object} rendered entirely in {hex}

20. A {object} designed in {hex} color

21. A realistic {hex}-colored {object}

22. A highly detailed {object} in hex {hex}

23. A {object} in rgb({r}, {g}, {b})

24. A {object} with the color rgb({r}, {g}, {b})

25. A {object} rendered in RGB color rgb({r}, {g}, {b})

26. A photo of a {object} in color rgb({r}, {g}, {b})

27. A {object} with color rgb({r}, {g}, {b})

### D.2 LIST OF CONTEXTUAL PROMPT TEMPLATES.

1. A {color} apple on a white plate

2. A {color} banana next to a sliced orange

3. A {color} carrot placed on a kitchen counter

4. A {color} mango in a fruit bowl with a lemon

5. A {color} strawberry on top of a dessert plate

6. A {color} broccoli beside a cutting board

7. A {color} guava resting in a wire fruit basket

8. A {color} papaya cut in half on a wooden table

9. A {color} lemon on a breakfast tray

10. A {color} car parked near a sidewalk

11. A {color} truck beside a loading dock

12. A {color} bus at a bus stop

13. A {color} motorcycle on a street corner

14. A {color} taxi in front of a building

15. A {color} jeep driving along a dirt road

16. A {color} sports car on a highway

17. A {color} train at a rural station

18. A {color} ferry approaching the dock

19. A {color} airplane at the runway gate

20. A {color} chair next to a wooden table

21. A {color} couch in front of a window

22. A {color} potted plant on a bookshelf

23. A {color} teapot on a breakfast tray

24. A {color} clock on a white wall

25. A {color} vase placed on a dining table

26. A {color} mug on a desk with books

27. A {color} candle beside a mirror

28. A {color} wardrobe beside a small chair

29. A {color} sink installed in a marble countertop

30. A {color} cat sleeping on a couch

31. A {color} dog playing with a ball

32. A {color} horse standing in a stable

33. A {color} sheep grazing on a green field

34. A {color} cow near a wooden fence

35. A {color} tiger behind a jungle bush

36. A {color} parrot on a tree branch

37. A {color} duck floating on a pond

38. A {color} owl perched on a wooden stump

39. A {color} goldfish swimming in a small tank

40. A {color} T-shirt folded on a table

41. A {color} jacket hanging on a coat rack

42. A {color} pair of jeans on a bed

43. A {color} hat resting on a chair

44. A {color} tie draped over a hanger

45. A {color} coat hanging near the door

46. A {color} backpack leaning against the wall

47. A {color} handbag on a desk

48. A {color} sports ball on a gym floor

49. A {color} kite flying in a clear sky

50. A {color} baseball glove on a bench

51. A {color} frisbee lying on the grass

52. A {color} snowboard resting against a wall

53. A {color} teddy bear on a child's bed

54. A {color} boxing glove placed on a shelf

55. A {color} doll sitting in a toy stroller

56. A {color} microwave on a kitchen shelf

57. A {color} hair dryer on a bathroom counter

58. A {color} toaster beside a coffee machine

59. A {color} refrigerator in the corner of the kitchen

60. A {color} cutting board on a kitchen island

61. A {color} sponge near a faucet

62. A {color} ruler beside an open notebook

63. A {color} fan placed near a window

### D.3   LIST OF SCENE DESCRIPTIVE PROMPT TEMPLATES.

1. A {color1} banana and a {color2} apple on a wooden table

2. A {color1} dog next to a {color2} cat and a {color3} couch in a living room

3. A {color1} skateboard beside a {color2} sports ball and a {color3} baseball bat

4. A {color1} jeep parked near a {color2} ambulance on a rainy street

5. A {color1} T-shirt and a {color2} pair of jeans folded on a bed

6. A {color1} microwave next to a {color2} refrigerator and a {color3} toaster

7. A {color1} tie hanging beside a {color2} hat and a {color3} jacket

8. A {color1} zebra standing with a {color2} giraffe in the savannah

9. A {color1} teddy bear and a {color2} doll placed on a shelf

10. A {color1} papaya, a {color2} guava, and a {color3} lemon in a fruit basket

11. A {color1} goldfish swimming with a {color2} turtle and a {color3} shark

12. A {color1} chair and a {color2} desk in a sunlit room

13. A {color1} surfboard, a {color2} kite, and a {color3} frisbee on the beach

14. A {color1} oven and a {color2} sink in a small kitchen

15. A {color1} cow grazing with a {color2} horse in a green field

16. A {color1} boat and a {color2} ferry docked at the harbor

17. A {color1} wardrobe and a {color2} bookcase against a blue wall

18. A {color1} duck floating near a {color2} parrot perched on a tree

19. A {color1} fan, a {color2} computer mouse, and a {color3} cutting board on the table

20. A {color1} umbrella leaning against a {color2} suitcase

21. A {color1} couch with a {color2} potted plant beside it

22. A {color1} car parked near a {color2} bus and a {color3} truck

23. A {color1} bear standing near a {color2} elephant in the wild

24. A {color1} book and a {color2} clock on a wooden shelf

25. A {color1} balloon tied to a {color2} snowboard and a {color3} boxing glove

26. A {color1} sink and a {color2} hair dryer on a bathroom counter

27. A {color1} strawberry and a {color2} mango next to a {color3} orange

28. A {color1} tie and a {color2} backpack on a desk

29. A {color1} shark chasing a {color2} lobster underwater

30. A {color1} remote beside a {color2} mug and a {color3} candle

## D.4 Implicit Color Association Prompt Templates

1. A {color} backpack is placed next to a suitcase that has the same color as the backpack, making their colors clearly match

2. A bicycle painted in {color} is parked beside a car that shares this exact color, making their similarity obvious A {color} dog is sitting near a cat whose fur matches the dog's color perfectly.

3. A {color} chair is positioned close to a couch that is painted in the same color, showing clear color similarity

4. An airplane painted {color} flies above a bus that is painted the same color, making the matching colors easy to notice

5. A tie colored in {color} lies beside a handbag that has matching color accents, clearly showing their shared color A {color} banana rests next to an apple that displays the same color as the banana's peel.

6. A motorcycle painted {color} is parked next to a truck sharing the exact same color, making their colors clearly identical A {color} goldfish is swimming near a turtle whose shell shows a similar color pattern.

7. A carrot with a {color} surface lies close to an orange that shares the same color, making the resemblance clear

8. A {color} baseball bat is resting against a skateboard that has the same color as the bat, showing a perfect match An elephant painted {color} is standing near a giraffe whose colors closely resemble the elephant's.

9. A {color} book is placed beside a clock that shares the same color, making it easy to see their matching appearance

10. A refrigerator painted {color} stands next to an oven that has been painted in the same color, clearly matching each other

11. A hat colored {color} rests on a jacket of identical color, making their shared shade obvious

12. A frisbee flying in {color} is near a kite that has the same color, making their similarity clear A sports ball painted {color} lies next to a baseball glove that shares the same color.

13. A handbag colored {color} hangs near an umbrella with matching color tones, clearly showing they share the same color A couch painted {color} is set beside a potted plant whose color scheme matches the couch's perfectly.

14. A train painted {color} is passing near a taxi painted in the same color, making their matching colors easy to identify

15. A microwave colored {color} is placed on a counter next to a toaster with the same color, showing clear color correspondence

16. A football helmet painted {color} lies next to a boxing glove with matching color, making their similarity obvious

17. A table painted {color} is set near a candle that shares the same color, making their resemblance clear A pair of jeans in {color} is folded next to pants that have the exact same color.

18. A car painted {color} is parked beside a minivan sharing the same color, making their colors clearly match

19. A tiger with {color} stripes is standing near a crocodile that has similar color tones, showing clear color resemblance

20. A toy terrier colored {color} sits close to a toy poodle that shares the same color, making their colors identical

21. A knife painted {color} is lying on a cutting board with matching color, showing a clear color match

22. A dog with {color} fur stands beside a horse that has the same color, making the shared color obvious

23. A suitcase colored {color} is placed next to a backpack of identical color, making their matching colors clear

24. A mug painted {color} is sitting next to a teapot with the same color, clearly matching each other

25. A car in {color} is parked beside a sports car of the same color, showing clear color similarity

26. An owl colored {color} is perched near a parrot sharing the same color, making the color match easy to see

27. A hair dryer painted {color} lies close to a remote control with matching color, clearly showing their similarity

28. A frisbee colored {color} is placed near a surfboard of the same color, showing clear color correspondence

29. An elephant painted {color} is standing next to a bear that has matching color, making the colors clearly match

30. A sweatshirt in {color} is folded beside a jacket that shares the same color, making their colors clearly identical

31. An apple colored {color} lies beside an orange that has the same color, making their similarity obvious

32. An airplane painted {color} is flying above a bus painted in the same color, showing a clear match

33. A skateboard colored {color} is lying near a pair of skis sharing the same color, making their colors match perfectly

34. A computer mouse colored {color} is lying near a cutting board that has the same color, making the color similarity clear

35. A banana with {color} peel is placed beside a mango that shares the same color, making their matching colors obvious

36. A sports ball colored {color} is lying near a football helmet with the same color, showing clear color similarity

37. A suitcase painted {color} is standing next to an umbrella of the same color, making the color match easy to see

38. A bear colored {color} is standing near a zebra that has matching color patterns, showing clear color resemblance

39. A remote control painted {color} is resting on a microwave that shares the same color, making the match clear

40. A dog with {color} fur is sitting next to a cat with identical color fur, making their colors obviously the same

41. An airplane painted {color} is flying above a truck painted the same color, making the color match obvious

42. A baseball bat colored {color} is lying near a frisbee of matching color, clearly showing their color similarity

43. A chair painted {color} is placed next to a table that has the same color, making the matching colors easy to see

44. A jacket colored {color} is hanging beside a coat of identical color, making their shared color obvious

45. A handbag painted {color} is resting on a backpack sharing the same color, clearly showing their color similarity

46. A car painted {color} is parked near a jeep painted the same color, making the matching colors obvious

47. A clock colored {color} is hanging near a vase of matching color, showing clear color correspondence

48. A crocodile colored {color} is swimming close to a shark with similar color, making their colors clearly alike

49. A toaster painted {color} is placed beside a refrigerator with matching color, showing their clear color match

50. A dog with {color} fur is standing beside a horse that has the same color, making the color similarity obvious

51. A sports ball painted {color} is lying near a golf ball with matching color, showing clear color resemblance

52. An umbrella colored {color} is hanging near a suitcase of the same color, making the colors clearly match

53. A chair painted {color} is placed next to a couch of identical color, showing their matching colors clearly

54. A baseball glove colored {color} is lying beside a baseball bat with matching color, making the color similarity obvious

55. A mango colored {color} is placed next to a papaya of the same color, showing clear color matching

56. A tie colored {color} is lying on a jacket with matching color, clearly showing their color similarity

57. A cat with {color} fur is sitting next to a dog with the same color fur, making the shared color obvious

58. A surfboard painted {color} is lying near a skateboard with identical color, making their colors clearly the same

59. A chair painted {color} is placed next to a desk with matching color, showing a clear color match

60. A microwave painted {color} is standing next to an oven of matching color, making the color similarity obvious

61. A baseball bat colored {color} is lying near a kite with the same color, clearly showing their matching colors

62. A bicycle painted {color} is parked beside a motorcycle sharing the same color, making their colors clearly alike

63. A parrot colored {color} is perched close to an owl with matching color, showing their shared color clearly

64. A dog with {color} fur is sitting beside a teddy bear of the same color, making the color match obvious

65. A football helmet painted {color} is lying near a snowboard with matching color, showing clear color similarity

66. An apple colored {color} is placed next to a guava of the same color, making the matching colors easy to see

67. A chair painted {color} is placed near a bookcase with matching color, making their colors clearly match

68. A suitcase colored {color} is resting next to a handbag of identical color, showing clear color correspondence

69. An orange colored {color} is lying near a carrot with the same color, making their colors clearly alike

70. A refrigerator painted {color} is standing next to a toaster with matching color, showing clear color similarity

71. A horse colored {color} is standing beside a sheep sharing the same color, making the color match obvious

72. A sports ball painted {color} is lying near a football helmet with matching color, showing clear color resemblance

73. A handbag colored {color} is placed next to a coat of the same color, making the colors clearly the same

74. A baseball bat colored {color} is lying near a sports ball of matching color, showing clear color similarity

75. A car painted {color} is parked beside a taxi painted the same color, making the colors clearly match

76. A clock colored {color} is hanging near a vase of matching color, showing clear color correspondence

77. A tie colored {color} is lying near a pair of pants with the same color, making the matching colors obvious

78. A dog with {color} fur is sitting beside a cat with identical color fur, showing their matching colors clearly

79. A baseball glove colored {color} is lying near a baseball bat with matching color, making their color similarity clear

80. A motorcycle painted {color} is parked next to a truck sharing the same color, making the color match obvious

81. An apple colored {color} is lying beside an orange of the same color, showing their matching colors clearly

82. A sweatshirt colored {color} is lying beside a pair of jeans with matching color, making their colors clearly the same

83. A remote painted {color} is resting on a microwave of the same color, making their color similarity clear

84. An umbrella colored {color} is hanging near a backpack with matching color, showing clear color correspondence

85. A banana colored {color} is placed next to a mango of the same color, making their colors clearly alike

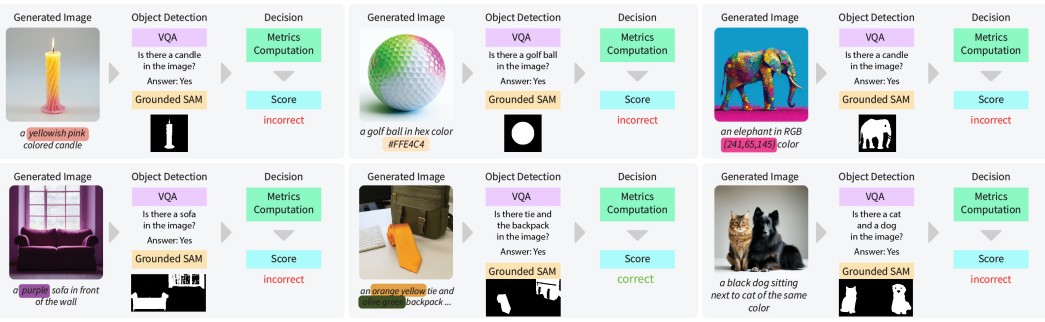

Figure 1: Qualitative Examples of the Color Generations of T2I models for different prompt settings.

| CSS3/X11 Color Space (2,058 samples, 6 questions) | | | | | | | | | |
|---|---|---|---|---|---|---|---|---|---|
| Model | Overall | Open-Ended | MCQ | Binary | Q1: Name | Q2: MCQ | Q3: Hex | Q4: Name Ver. | Q5: RGB Ver. | Q6: Hex Ver. |
| Janus | 26.56 | 5.03 | 12.20 | 52.44 | 7.34 | 12.20 | 2.72 | 57.43 | 50.00 | 49.90 |
| Janus Pro | 28.64 | 6.63 | 19.44 | 50.95 | 7.48 | 19.44 | 5.78 | 55.00 | 47.18 | 50.68 |
| BLIP3o | 31.41 | 12.46 | 24.73 | 57.08 | 9.96 | 24.73 | 14.38 | 60.50 | 57.05 | 53.69 |
| Deepseek | 28.72 | 11.35 | 18.85 | 55.85 | 9.43 | 18.85 | 13.27 | 56.41 | 53.60 | 57.53 |
| Qwen2-VL | 29.13 | 9.36 | 23.23 | 54.81 | 8.94 | 23.23 | 9.77 | 55.20 | 53.21 | 56.03 |
| Instruct-VL | 27.96 | 7.19 | 20.55 | 56.17 | 8.94 | 20.55 | 5.44 | 56.75 | 55.00 | 56.75 |
| mPLUG-Owl3 | 26.76 | 7.24 | 17.93 | 55.00 | 6.95 | 17.93 | 7.53 | 56.41 | 54.37 | 54.23 |

| IBCC L2 Color Space (406 samples, 4 questions) | | | | | | | |
|---|---|---|---|---|---|---|---|
| Model | Overall | Open-Ended | MCQ | Binary | Q1: Name | Q2: MCQ | Q3: Name Ver. | Q4: RGB Ver. |
| Janus | 37.68 | 25.86 | 33.99 | 55.79 | 25.86 | 33.99 | 61.58 | 50.00 |
| Janus Pro | 44.02 | 26.60 | 43.60 | 55.91 | 26.60 | 43.60 | 64.53 | 47.29 |
| BLIP3o | 45.13 | 25.12 | 45.81 | 64.41 | 25.12 | 45.81 | 72.41 | 56.40 |
| Deepseek | 45.50 | 27.34 | 45.32 | 63.30 | 27.34 | 45.32 | 72.91 | 53.69 |
| Qwen2-VL | 45.16 | 24.63 | 43.35 | 63.67 | 24.63 | 43.35 | 69.46 | 57.88 |
| Instruct-VL | 44.91 | 26.60 | 45.57 | 62.56 | 26.60 | 45.57 | 72.41 | 52.71 |
| mPLUG-Owl3 | 44.95 | 24.14 | 42.12 | 62.57 | 24.14 | 42.12 | 72.91 | 52.22 |

Table 6: Comprehensive Performance Analysis of VLM-based VQA on ISCC NBS Level 2 and CSS3/X11 colors.

86. A knife colored {color} is lying on a cutting board with matching color, showing clear color similarity

87. A dog with {color} fur is standing near a horse with identical color, making the matching colors obvious

88. A tie colored {color} is lying over a jacket of the same color, making their colors clearly the same

89. A surfboard colored {color} is lying near a skateboard sharing the same color, showing clear color resemblance

90. A handbag painted {color} is resting on a suitcase with matching color, making the color match obvious

91. A chair colored {color} is placed next to a couch of identical color, showing clear color similarity

92. A football helmet painted {color} is lying near a boxing glove with the same color, making their colors clearly the same

93. A tiger colored {color} is standing close to a crocodile sharing the same color, showing clear color correspondence

# E    LIMITATIONS OF VQA-BASED ASSESSMENT METHODS

Table 6 presents a detailed evaluation of three visual-language models—Janus, Janus Pro, and mPLUG-large—on color understanding tasks grounded in two standard color sets: CSS3/X11 and ISCC-NBS Level 2. The goal of this benchmark is to probe how well VLMs can interpret, reason about, and verify colors in images using natural language.

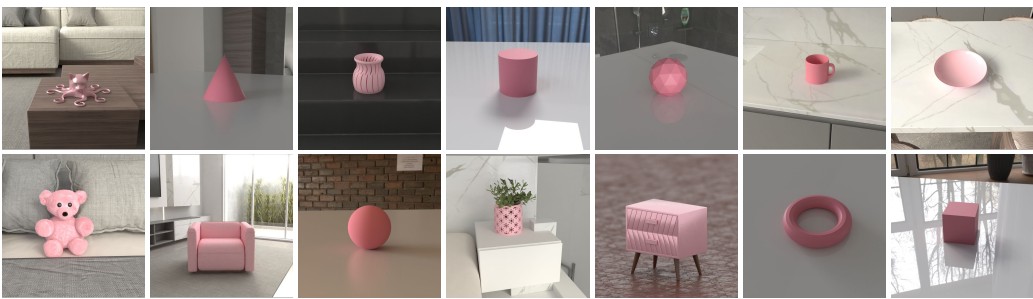

Figure 2: Set of 14 objects rendered in blender in CSS3/X11 colors for VQA evaluation.

We design a set of structured visual question answering (VQA) tasks, covering three reasoning modes: (1) Open-Ended Question, where the model must produce a color name or code in free-form; (2) Multiple Choice Question, where it must choose the correct answer from a list; and (3) Binary Question, where it must answer "Yes" or "No" to a color-specific query.

Each model is tested on a set of image-question pairs derived from real generations using known target colors. we render the 14 objects in the CSS3/X11 and ISCC-NBS L2 colors in blender, as shown in Figure. 2.

We evaluate six question types under the CSS3/X11 taxonomy (2058 samples), and four question types for ISCC-NBS Level 2 (406 samples), listed below:

**CSS3/X11 Questions:**

- Q1 (Generative): What is the CSS3/X11 color name of the {object} in the given image?
- Q2 (Discriminative): Given the list of CSS3/X11 color names [...], what is the color name of the {object} in the image?
- Q3 (Generative): What is the CSS3/X11 hex color code of the {object} in the image?
- Q4 (Verification): Is the color name of the {object} {color} in the image?
- Q5 (Verification): Is the RGB color of the {object} {r,g,b} in the image?
- Q6 (Verification): Is the hex color code of the {object} {hex code} in the image?

**ISCC-NBS Level 2 Questions:**

- Q1 (Generative): What is the IBCC Level 2 color name of the {object} in the image?
- Q2 (Discriminative): Given the list of IBCC Level 2 color names [...], what is the color name of the {object}?
- Q3 (Verification): Is the color name of the {object} {color} in the image?
- Q4 (Verification): Is the RGB color of the {object} {r,g,b} in the image?

From the results presented in Table 6, we observe that binary verification tasks consistently yield the highest accuracy, while open-ended generative tasks are particularly weak across all models and color spaces. For instance, in the CSS3/X11 benchmark, open-ended scores (Q1 and Q3) remain below 8% across all models, with the best-performing model (BLIP3o) achieving only 12.46% in open-ended tasks overall. This suggests that models struggle to produce the correct color name or hex code when not explicitly prompted with options.

In contrast, binary verification tasks (Q4–Q6) see much higher accuracy—often above 50%—indicating that VLMs are better at recognizing and verifying predefined information than generating it. MCQ tasks (Q2) perform moderately well, achieving up to 24.73% (BLIP3o), but still depend on the presence of semantically close distractors and color naming consistency.

Similar trends appear in the IBCC L2 evaluation. Although the smaller color set improves overall performance, open-ended results remain limited (e.g., Deepseek: 27.34%), and binary verification continues to dominate (e.g., achieving up to 64.41% in BLIP3o and 63.67% in Qwen2-VL).

| Model | Color Type | Clothes_Acc | Vehicle | Furniture_House | Tools_Misc | Sports_Toys | Animals | Fruit_Veg |
|---|---|---|---|---|---|---|---|---|
| Flux | css | 39.5554 | 39.4843 | 37.4237 | 35.9550 | 34.9350 | 30.5794 | 21.2568 |
| Flux | l2 | 60.4044 | 56.9160 | 51.2754 | 50.3166 | 49.2966 | 41.9628 | 37.4442 |
| Flux | l3 | 51.9078 | 49.9698 | 45.2676 | 43.8498 | 43.2990 | 38.4948 | 29.6718 |
| Flux | overall | 50.6226 | 48.7899 | 44.6556 | 43.3737 | 31.0616 | 37.0124 | 29.4576 |
| PixArt-Alpha | css | 41.3814 | 50.4084 | 44.8596 | 45.8796 | 44.4516 | 40.6266 | 24.2250 |
| PixArt-Alpha | l2 | 62.1486 | 56.4468 | 64.3722 | 56.7834 | 58.2930 | 55.8348 | 38.8824 |
| PixArt-Alpha | l3 | 56.5794 | 46.6446 | 55.3962 | 48.7866 | 52.5402 | 45.9816 | 31.2018 |
| PixArt-Alpha | overall | 53.3700 | 51.1666 | 54.8760 | 50.4832 | 51.7616 | 47.4810 | 31.4364 |
| PixArt-Sigma | css | 37.8318 | 42.7380 | 42.4932 | 42.6156 | 44.5128 | 37.9338 | 29.2332 |
| PixArt-Sigma | l2 | 58.7520 | 51.9588 | 62.5668 | 57.0048 | 58.8744 | 47.7972 | 46.9098 |
| PixArt-Sigma | l3 | 55.6818 | 43.2480 | 53.6724 | 46.8078 | 52.6320 | 38.0256 | 34.1086 |
| PixArt-Sigma | overall | 50.7552 | 45.9816 | 52.9108 | 48.8036 | 52.0064 | 41.2522 | 36.7472 |
| Sana | css | 46.1550 | 50.5818 | 44.2068 | 47.5728 | 51.4794 | 48.3378 | 23.3376 |
| Sana | l2 | 62.3730 | 50.7960 | 58.4154 | 57.4974 | 60.5370 | 58.1298 | 40.2186 |
| Sana | l3 | 58.1706 | 45.6144 | 53.1012 | 49.2354 | 54.6312 | 50.0514 | 22.0422 |
| Sana | overall | 55.5662 | 48.9974 | 51.9078 | 51.4352 | 55.5492 | 52.1730 | 28.5328 |
| SD 3.5 | css | 42.7176 | 45.4920 | 41.8506 | 44.5536 | 38.0460 | 38.0154 | 22.3788 |
| SD 3.5 | l2 | 64.0458 | 59.2926 | 61.9848 | 61.3530 | 56.4468 | 52.0812 | 39.0660 |
| SD 3.5 | l3 | 62.1384 | 52.9686 | 56.7630 | 51.6936 | 53.1624 | 48.9804 | 28.4478 |
| SD 3.5 | overall | 56.3006 | 52.5844 | 53.5466 | 52.5334 | 49.2184 | 46.3590 | 30.0642 |
| SD 3 | css | 42.5034 | 43.7070 | 41.8000 | 42.7788 | 41.4936 | 40.6062 | 25.0716 |
| SD 3 | l2 | 55.4370 | 48.6438 | 56.5386 | 51.0306 | 44.8596 | 49.7964 | 37.1178 |
| SD 3 | l3 | 54.0030 | 48.9804 | 52.9980 | 52.8360 | 42.0444 | 47.6646 | 32.0586 |
| SD 3 | overall | 50.6430 | 47.1104 | 50.4594 | 48.8818 | 42.7992 | 45.9924 | 31.4160 |
| Janus-Pro | css | 24.0822 | 25.4490 | 28.0602 | 25.4490 | 25.2246 | 13.1886 | 7.8642 |
| Janus-Pro | l2 | 46.4100 | 35.6898 | 41.8914 | 34.4862 | 39.2600 | 26.1834 | 19.2576 |
| Janus-Pro | l3 | 42.8400 | 31.0794 | 39.9330 | 32.3442 | 35.6490 | 24.3474 | 16.1772 |
| Janus-Pro | overall | 37.7774 | 30.7394 | 36.6282 | 30.7600 | 33.3778 | 21.2398 | 14.4330 |
| OmniGen2 | css | 35.4246 | 34.0884 | 36.3324 | 39.6270 | 41.3508 | 27.2034 | 23.6334 |
| OmniGen2 | l2 | 60.5472 | 46.1856 | 60.7308 | 54.3660 | 55.8144 | 41.3814 | 38.0766 |
| OmniGen2 | l3 | 48.3886 | 38.8416 | 49.0314 | 45.0534 | 47.0120 | 34.8024 | 28.4580 |
| OmniGen2 | overall | 48.1168 | 39.7052 | 48.6982 | 46.3488 | 48.0488 | 34.4624 | 30.0560 |
| BLIP3o | css | 34.1496 | 35.1696 | 33.6192 | 43.8294 | 42.1464 | 24.6228 | 28.8048 |
| BLIP3o | l2 | 47.9706 | 42.2076 | 45.9204 | 54.5496 | 52.3464 | 33.8130 | 42.4218 |
| BLIP3o | l3 | 46.6140 | 37.3932 | 37.5462 | 49.0518 | 43.5540 | 30.0180 | 34.8942 |
| BLIP3o | overall | 42.9114 | 38.2568 | 39.0286 | 49.1436 | 45.9956 | 29.4678 | 35.3736 |

Table 7: Per-category performance across all models and scoring types (css, l2, l3, overall).

These results highlight a critical limitation that VLMs are not reliable for evaluating color fidelity. First, their low open-ended accuracy indicates poor internal representation of precise color semantics despite correct visual grounding. Second, their strong binary verification performance may result from learning patterns in color datasets rather than actual understanding. Third, performance varies significantly across color taxonomies and question types, exposing instability in color reasoning.

Ultimately, while VLMs can aid in approximate visual understanding, they are not suitable as color evaluation agents in benchmarks requiring accurate color reproduction. This motivates the need for dedicated, metric-based evaluation pipeline rather than relying on VQA responses for color assessment.

# F  CATEGORY-WISE QUALITATIVE ANALYSIS OF T2I MODELS

Table 7 reveals strong evidence of model bias and entanglement between object semantics and color generation. Across all the employed models, performance varies significantly depending on both the object category and the chosen color taxonomy. For example, the models perform better on categories like Clothing, Furniture, and Vehicles, while they struggle on categories like Fruits and Vegetables and Animals. This discrepancy indicates that models are not disentangling color prompts from natural objects training priors. Instead, they tend to reproduce default or canonical colors seen during training (e.g., red apples, yellow bananas), even when explicitly instructed otherwise. Some examples are shown in the Figure 1.

Furthermore, the models perform markedly better with coarse-grained color taxonomies like ISCC-NBS Level 2, but struggle with the finer granularity of CSS3 and Level 3, indicating a lack of standard color understanding. The combination of these effects highlights that the current T2I models often conflate object identity with memorized color distributions, failing to generalize to atypical or prompt-specified colors. This entanglement points to a systemic bias in training data and architecture, which limits their usefulness for tasks requiring precise and independent control over object appearance attributes—such as color.

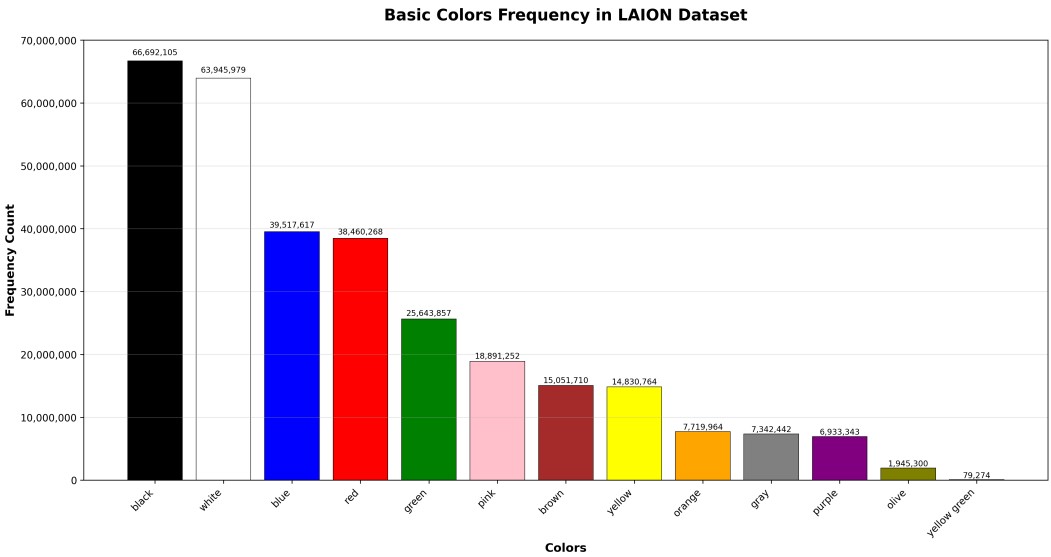

Figure 3: Frequency Analysis of Basic Colors in LAION-2B text prompts.

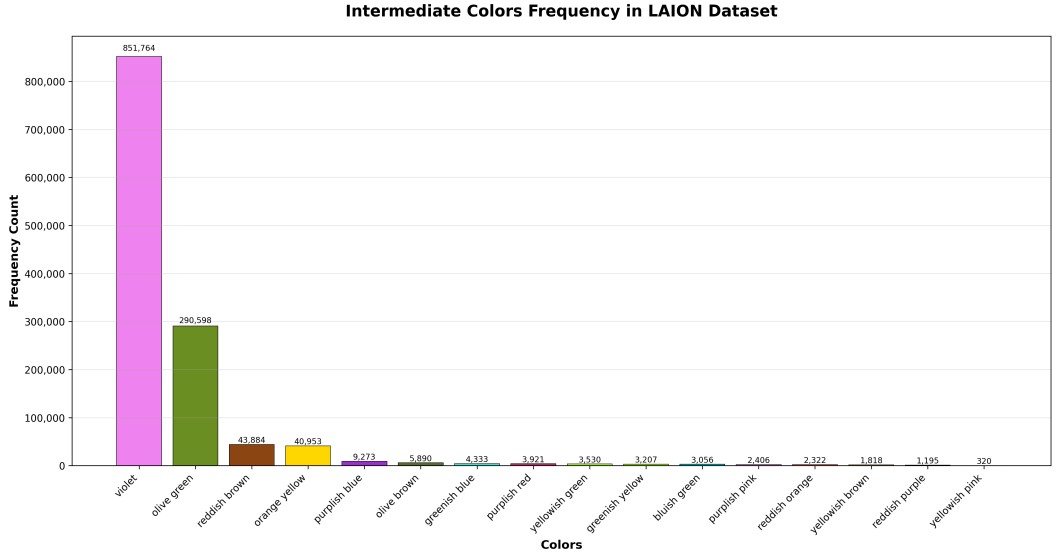

Figure 4: Frequency Analysis of Intermediate Colors in LAION-2B text prompts.

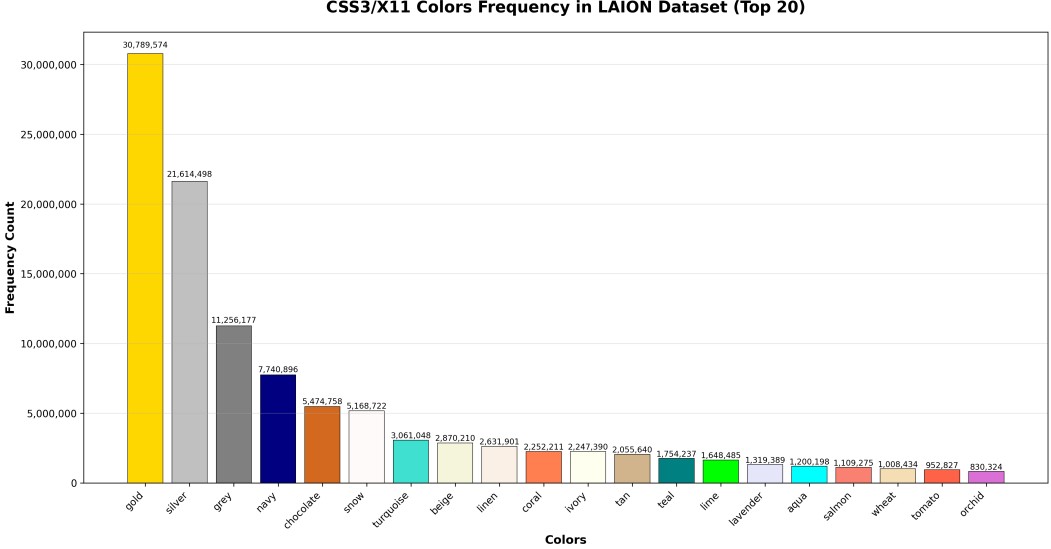

Figure 5: Frequency Analysis of CSS3/X11 Colors in LAION-2B text prompts.

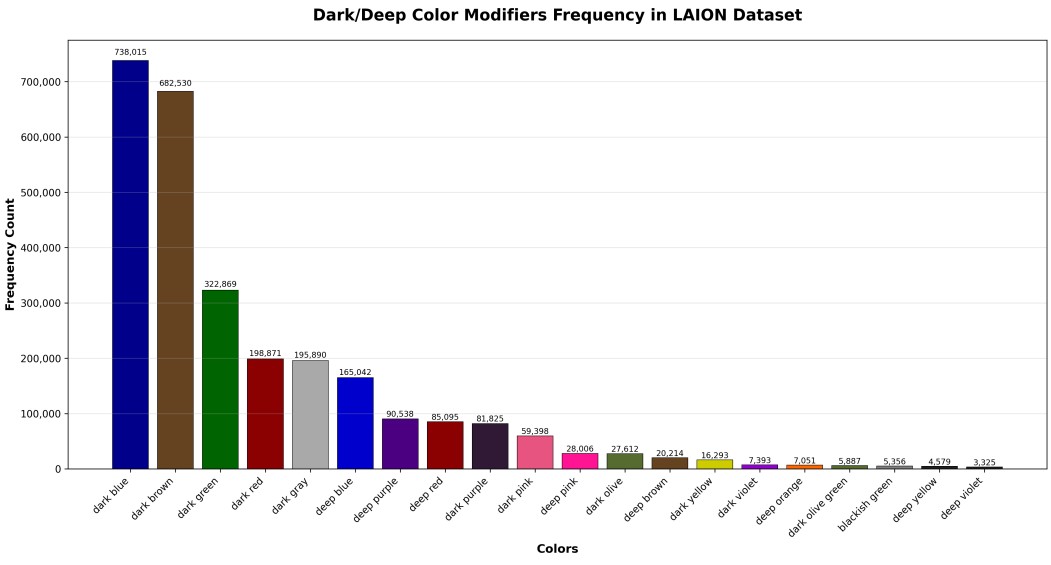

Figure 6: Frequency Analysis of Dark Color Modifiers in LAION-2B text prompts.

Figure 7: Frequency Analysis of Light Color Modifiers in LAION-2B text prompts.