# OpenReview forum: "GenColorBench: A Color Evaluation Benchmark for Text-to-Image Generation Models"
_ICLR.cc/2026/Conference — ICLR 2026 Conference Withdrawn Submission_

### Official Review · Reviewer_v14i · 2025-10-29

**Soundness:** 3
**Presentation:** 3
**Contribution:** 3
**Rating:** 6
**Confidence:** 3

**Summary:**

This paper proposes a benchmark for evaluating the capability of T2I models in faithfully rendering the specified colors on generated objects.

**Strengths:**

1. To the best of my knowledge, color evaluation is indeed an overlooked aspect in existing T2I benchmarks, and this work therefore fills an important gap.
2. The construction method, and especially the color identification protocol (lines 288-314), seems well-thought-out. However, as I am not an expert in color systems, I am not in a position to judge the reasonability, correctness, and professionalism of this specific design.
3. The benchmark covers multiple dimensions of evaluation, which I find to be reasonable and useful.
4. The findings in Section 4 are informative. While some of these findings might seem straightforward or have been observed in prior works, it is highly valuable that the proposed benchmark allows these informal observations to be quantitatively validated.

**Weaknesses:**

1. I suggest designing a hierarchy of evaluation protocols with increasingly fine-grained color divisions. At a minimum, I would recommend adding a protocol that only involves ISCC-NBS Level 1 color names. The underlying rationale is that highly fine-grained color specification currently seems to be a niche demand, and such evaluation might be more relevant for specialized models. Subjecting general-purpose models to such strict criteria may not be necessary. A protocol with a coarser color division, perhaps coupled with greater tolerance, could provide more instructive scores for these models.
2. (This is a suggestion, not a mandatory requirement.) I would encourage the authors to perform a correlation analysis between the models' scores on GenColorBench and their scores on other benchmarks, such as GenEval or T2I-CompBench. Such an analysis could help derive interesting conclusions, for example, whether models with strong compositional generation or instruction-following capabilities necessarily exhibit stronger color-adherence abilities.

**Questions:**

None

---

### Official Review · Reviewer_wWRc · 2025-11-01

**Soundness:** 3
**Presentation:** 3
**Contribution:** 2
**Rating:** 4
**Confidence:** 4

**Summary:**

This paper introduces GenColorBench, the first large-scale benchmark explicitly designed to evaluate color generation accuracy and understanding in text-to-image (T2I) and unified multimodal generative models. It identifies a gap in current evaluation benchmarks (e.g., GenEval, T2I-CompBench++, TIFA) which assess compositionality, faithfulness, or reasoning but ignore fine-grained color controllability.

- Comprehensive Design:
44K+ prompts across 5 tasks; supports linguistic and numerical color representations; standardized with ISCC–NBS and CSS3/X11 mappings.

- Evaluation Protocol:
Introduces a hybrid perceptual–metric approach using CIELAB color space, PCA-based dominant hue extraction, and multiple JND-thresholded metrics.

- Empirical Insights:
Evaluation of major diffusion, autoregressive, and multimodal models (e.g., Flux, SD3.5, BLIP3o, OmniGen2) reveals:
Models perform well on basic colors but fail on intermediate or modifier-based colors (e.g., greenish blue).
Color–object association and multi-object composition remain unsolved.
Numeric color control (RGB/hex) is particularly poor across models.

**Strengths:**

1. The paper presents a clear idea, addressing an interesting aspect of text-to-image evaluation — color understanding.
2. The writing is generally clear and structured, with sufficient methodological detail and logical flow.
3. The work offers a novel a benchmark contribution, supported by sound experimental design and comprehensive analysis across multiple models.

**Weaknesses:**

1. Each object is evaluated with only a single dominant color, which may oversimplify real-world cases where objects naturally exhibit multiple colors or textures.

2. There is some concern about the practical relevance of the benchmark—generative models may not need to distinguish over 400 colors, many of which are not practical or barely perceptible even to humans.

**Questions:**

- 1. I find the description of object colors in the benchmark somewhat confusing. In reality, many objects naturally have multiple colors. For instance, in Figure 1, the pink car contains several other colored parts, and the “yellow cat” clearly shows a mix of yellow and white. How does the benchmark handle single objects that contain multiple colors? Has any evaluation been conducted for such cases? How to select the dominant color?


- 2. Quality control is crucial for a benchmark. However, the paper provides insufficient discussion regarding human quality control. Human judgments of color can be subjective and imprecise. The authors should clarify how they address these issues, as well as important aspects such as the proportion of data retained after manual quality inspection.

- 3. I’d really appreciate it if you could replay to the two concerns mentioned in the “Weaknesses” section.

---

### Official Review · Reviewer_9ap3 · 2025-11-01

**Soundness:** 2
**Presentation:** 4
**Contribution:** 4
**Rating:** 6
**Confidence:** 4

**Summary:**

This paper introduces GenColorBench, the first large-scale benchmark designed to systematically evaluate color generation capabilities in text-to-image (T2I) models.
The authors identify a critical gap, as existing benchmarks often overlook color precision.
GenColorBench features over 44,000 prompts grounded in established color science systems (ISCC-NBS, CSS3/X11) and uniquely incorporates numerical color formats (RGB/hex).
It proposes a multi-faceted evaluation framework across five distinct tasks, using a sophisticated pipeline involving VQA for object detection, Grounded SAM for segmentation, and a perceptually-grounded metric in CIELAB space to score color accuracy.
The extensive evaluation of state-of-the-art models reveals profound and widespread difficulties in achieving precise color control, particularly for complex prompts, and suggests that models rely heavily on dataset biases rather than true compositional understanding.
The work provides a valuable resource and a strong baseline for future research on color fidelity in T2I generation.

**Strengths:**

- Addresses a Critical Gap:
The paper tackles a well-motivated and highly important limitation in current T2I evaluation.
Precise color control is a fundamental requirement for many practical applications, and this work provides the first systematic, large-scale tool to measure it.
- Theoretically Grounded Methodology:
The benchmark's design is well-founded in color science.
Grounding the evaluation in established, perceptually uniform color systems like ISCC-NBS and employing the 'dominant hue' concept for analysis represents a significant advance over simplistic VQA or CLIP-based scoring.
- Comprehensive Scale and Coverage:
With over 44,000 prompts, 400+ colors, and five distinct evaluation tasks (including the novel Numerical Color Understanding task), the benchmark provides unprecedented scale and specificity, enabling a deep and nuanced assessment of model capabilities.
- Insightful Empirical Findings:
The paper does more than just introduce a benchmark; it uses it to deliver key insights.
The findings—that all models struggle significantly, that performance is tied to semantic categories, and that models reflect training data biases—are valuable contributions to the field's understanding of current model limitations.
- Exceptional Clarity and Presentation:
The paper is exceptionally well-written, logically structured, and easy to understand.
The motivation is clear, and the figures and tables (especially Table 1) are highly effective at communicating the core contributions and results.

**Weaknesses:**

- Benchmark Calibration Concerns:
The performance scores across all evaluated models are extremely low (highest average is 22.42%).
Without a human performance baseline or inter-annotator agreement study, it is difficult to ascertain whether these scores reflect genuine, severe model limitations or overly stringent evaluation criteria.
This lack of calibration makes the absolute scores hard to interpret.
- Arbitrary Thresholding in Evaluation Metric:
The Just-Noticeable-Difference (JND) threshold, a critical parameter in the scoring mechanism, is stated as 'typically 5' without strong justification.
The paper lacks a sensitivity analysis for this parameter, which could significantly impact the final results and conclusions.
- Reliability of the Evaluation Pipeline:
The framework's accuracy depends on a chain of pre-trained models (Janus-1.3B for VQA, Grounded SAM for segmentation), which have their own failure modes.
The paper itself shows that VLLMs struggle with fine-grained color, raising concerns about the reliability of using one for object detection gating without a thorough analysis of potential error propagation.
- Limited Analysis of Failure Modes:
While the benchmark effectively reveals that models perform poorly on numerical color understanding (most scoring under 10%), there is insufficient analysis into *why* this task is so challenging or what specific aspects of RGB/hex processing are most problematic for current architectures.

---

- General Limitations:
    - The benchmark focuses on English-centric color names and systems. Its applicability to non-English languages or cultures with different color taxonomies is not discussed.
    - A discussion on the potential negative societal impacts is missing. For example, a benchmark that perfects model adherence to specific colors could be used for malicious purposes, such as flawlessly replicating copyrighted brand identities or creating more convincing disinformation.
    - The evaluation protocol does not explicitly control for hyperparameters like sampling steps or guidance scale across models, which could affect the fairness of the comparison.

**Questions:**

- Given the very low scores across all models, what do you consider a 'good' or 'reasonable' performance level on this benchmark?
Would you consider conducting a human evaluation study on a subset of the prompts to establish a performance baseline and help calibrate the benchmark's difficulty?
- Could you provide a justification for choosing a JND threshold of 5?
How sensitive are the reported model rankings and overall conclusions to variations in this threshold (e.g., changing it to 3 or 7)?
- How does the known unreliability of VQA models for fine-grained tasks impact the initial object detection step?
Could you perform an ablation study using a different, stronger VQA model—or a human-in-the-loop validation on a subset—to assess the impact of this 'gating' model choice on the final results?
- Your analysis shows BLIP30 significantly outperforms other models on numerical color understanding.
Do you have any hypotheses as to why its multimodal architecture might be better suited for this specific task compared to the other models tested?

**Details Of Ethics Concerns:**

A discussion on the potential negative societal impacts is missing. For example, a benchmark that perfects model adherence to specific colors could be used for malicious purposes, such as flawlessly replicating copyrighted brand identities or creating more convincing disinformation.

---

### Official Review · Reviewer_AJHA · 2025-11-02

**Soundness:** 1
**Presentation:** 2
**Contribution:** 1
**Rating:** 2
**Confidence:** 4

**Summary:**

The paper introduces GenColorBench, a large-scale benchmark (44K+ prompts) for evaluating the ability of text-to-image (T2I) models to generate precise colors. The benchmark is grounded in color systems like ISCC-NBS and CSS3/X11 and includes tasks for color name accuracy, object association, and, notably, numerical color understanding (e.g., RGB/hex codes). The authors first argue that VLM-based evaluators are unreliable (Table 2) and then propose a pixel-grounded evaluation pipeline using CIELAB color space and "dominant hue" extraction . The paper's main finding is that all current SOTA T2I models perform very poorly on these tasks , with the best average score being only 22.42% (Table 4).

**Strengths:**

- Thorough Data Curation: The paper is thorough in its creation of the benchmark dataset, drawing from established color systems and creating a large number of prompts (44,464).
- Perceptual Metric Choice: The choice to use CIELAB color space for evaluation  is sound, as it is more perceptually uniform than RGB.

**Weaknesses:**

- Marginal/Trivial Contribution: The paper's core premise is flawed. It focuses on a niche, unimportant problem (hyper-specific color accuracy). This is a solved problem at a "good enough" level for most applications, and this benchmark does not measure any deeper semantic capability.
- Flawed Methodology: The evaluation pipeline is fundamentally unsound. It relies on a VQA model (Janus-1.3B)  that the paper itself proves is unreliable (Table 2).
- Non-Transparent Pipeline: The methodology relies on "black box" components, such as a GPT-generated "negative label" list, which makes the results un-auditable and impossible to reproduce reliably.
- Poor Organization: The paper is poorly structured, making it difficult to read and assess.

**Questions:**

- The paper's central weakness is its triviality. Can the authors provide any evidence that a model's (in)ability to generate "moderate purplish pink"  has any correlation with its ability to perform more complex, meaningful tasks like compositional reasoning or instruction following?
- How can you justify using Janus-1.3B for object validation  when your own Table 2 shows this model has near-zero capability in related vision-language tasks? Please provide accuracy metrics for Janus-1.3B on the specific object presence task it is used for.
- How can this benchmark be considered reliable when it relies on a non-transparent, GPT-generated list of "negative labels"? This seems like a critical, un-audited component that directly affects the final score.

---

### Author Response · Authors · 2025-11-14

We appreciate the reviewers’ helpful feedback. Given the borderline scores, we’ve decided to withdraw the paper for now and use the feedback to make a stronger revision. Thanks again for your time and effort — it will really help us improve this benchmark!

---

### Note · Authors · 2025-11-14

I have read and agree with the venue's withdrawal policy on behalf of myself and my co-authors.